# *ketu* mutant mice uncover an essential meiotic function for the ancient RNA helicase YTHDC2

Devanshi Jain[1], M Rhyan Puno[2,3], Cem Meydan[4,5], Nathalie Lailler[6], Christopher E Mason[4,5,7], Christopher D Lima[2,3], Kathryn V Anderson[8], Scott Keeney[1,3]*

[1]Molecular Biology Program, Memorial Sloan Kettering Cancer Center, New York, United States; [2]Structural Biology Program, Memorial Sloan Kettering Cancer Center, New York, United States; [3]Howard Hughes Medical Institute, Memorial Sloan Kettering Cancer Center, New York, United States; [4]Department of Physiology and Biophysics, Weill Cornell Medicine, New York, United States; [5]The HRH Prince Alwaleed Bin Talal Bin Abdulaziz Alsaud Institute for Computational Biomedicine, Weill Cornell Medicine, New York, United States; [6]Integrated Genomics Operation, Memorial Sloan Kettering Cancer Center, New York, United States; [7]The Feil Family Brain and Mind Research Institute, Weill Cornell Medicine, New York, United States; [8]Developmental Biology Program, Memorial Sloan Kettering Cancer Center, New York, United States

*For correspondence:
s-keeney@ski.mskcc.org

Competing interests: The authors declare that no competing interests exist.

**Abstract** Mechanisms regulating mammalian meiotic progression are poorly understood. Here we identify mouse YTHDC2 as a critical component. A screen yielded a sterile mutant, 'ketu', caused by a *Ythdc2* missense mutation. Mutant germ cells enter meiosis but proceed prematurely to aberrant metaphase and apoptosis, and display defects in transitioning from spermatogonial to meiotic gene expression programs. *ketu* phenocopies mutants lacking MEIOC, a YTHDC2 partner. Consistent with roles in post-transcriptional regulation, YTHDC2 is cytoplasmic, has 3′→5′ RNA helicase activity in vitro, and has similarity within its YTH domain to an $N^6$-methyladenosine recognition pocket. Orthologs are present throughout metazoans, but are diverged in nematodes and, more dramatically, Drosophilidae, where Bgcn is descended from a *Ythdc2* gene duplication. We also uncover similarity between MEIOC and Bam, a Bgcn partner unique to schizophoran flies. We propose that regulation of gene expression by YTHDC2-MEIOC is an evolutionarily ancient strategy for controlling the germline transition into meiosis.
DOI: https://doi.org/10.7554/eLife.30919.001

## Introduction

Sexual reproduction requires formation of gametes with half the genome complement of the parent organism. The specialized cell division of meiosis achieves this genome reduction by appending two rounds of chromosome segregation to one round of DNA replication (*Page and Hawley, 2003*). Homologous maternal and paternal chromosomes segregate in the first meiotic division, then sister centromeres separate in the second. Prior to the first division, homologous chromosomes pair and recombine to form temporary connections that stabilize the chromosomes on the metaphase I spindle (*Page and Hawley, 2003*; *Hunter, 2007*). Errors in these processes can cause gametogenic failure and infertility, or yield aneuploid gametes that in turn lead to miscarriage or birth defects in offspring (*Hassold and Hunt, 2001*; *Sasaki et al., 2010*).

In well-studied metazoan species, meiosis occurs specifically in a dedicated germ cell lineage after a period of limited expansion via mitotic 'transit-amplifying' cell divisions (*de Rooij, 2001*; *Davies and Fuller, 2008*). The coordination of germline stem cell divisions with entry into meiosis and the subsequent progression of cells through meiotic divisions are tightly regulated (e.g., *Griswold, 2016*), but mechanisms underlying this regulation are not fully understood, particularly in mammals. And, more generally, the catalog of mammalian genes required for germ cell development, meiosis, and gametogenesis remains incomplete. In efforts to overcome this lack, we carried out a phenotype-based, random chemical mutagenesis screen to identify novel mouse meiotic mutants. One hit was a male-sterile mutant we named *rahu*, for 'recombination-affected with hypogonadism from under-populated testes' (*Jain et al., 2017*). This mutant is defective for the function of a rodent-specific DNA methyltransferase paralog, DNMT3C. Here, we describe a new mutant that we named *ketu*, for 'keen to exit meiosis leaving testes under-populated'. Ketu and Rahu are harbingers of misfortune in Vedic mythology.

*ketu* is a missense mutation in *Ythdc2* (YTH-domain containing 2), which encodes a protein with RNA helicase motifs and a YT521-B homology (YTH) RNA-binding domain (*Stoilov et al., 2002*; *Morohashi et al., 2011*). Recombinant YTHDC2 protein displays 3′→5′ RNA helicase activity and a solution structure of its YTH domain is consistent with direct recognition of $N^6$-methyladenosine-containing RNA. *Ythdc2^{ketu}* homozygotes are both male- and female-sterile. In the testis, mutant germ cells carry out an abortive attempt at meiosis: they express hallmark meiotic proteins and initiate recombination, but fail to fully extinguish the spermatogonial mitotic division program, proceed prematurely to an aberrant metaphase-like state, and undergo apoptosis. This phenotype is similar to mutants lacking MEIOC, a meiosis-specific protein that was recently shown to be a binding partner of YTHDC2 and that has been proposed to regulate male and female meiosis by controlling the stability of various mRNAs (*Abby et al., 2016*; *Soh et al., 2017*). Our results thus reveal an essential role for YTHDC2 in the germlines of male and female mice and show that YTHDC2 is an indispensable functional partner of MEIOC. Furthermore, phylogenetic studies demonstrate that the YTHDC2-MEIOC complex is an evolutionarily ancient factor, present in the last common ancestor (LCA) of Metazoa. Nevertheless, despite high conservation in most metazoans, we uncover unexpectedly complex evolutionary patterns for YTHDC2 and MEIOC family members in specific lineages, particularly nematodes and the Schizophora section of flies, which includes *Drosophila melanogaster*.

## Results

### Isolation of the novel meiotic mutant *ketu* from a forward genetic screen

To discover new meiotic genes, we carried out a phenotype-based, random mutagenesis screen in mice (*Jain et al., 2017*). Mutagenesis was performed by treatment of male mice of the C57BL/6J strain (B6 hereafter) with the alkylating agent *N*-ethyl-*N*-nitrosourea (ENU). ENU introduces de novo mutations in the germline, predominantly single base substitutions (*Hitotsumachi et al., 1985*; *Caspary and Anderson, 2006*; *Probst and Justice, 2010*). To uncover recessive mutations causing defects in male meiosis, we followed a three-generation breeding scheme including outcrossing with females of the FVB/NJ strain (FVB hereafter) (*Caspary and Anderson, 2006*; *Caspary, 2010*; *Jain et al., 2017*) (*Figure 1A*).

Third-generation (G3) male offspring were screened for meiotic defects by immunostaining squash preparations of testis cells for SYCP3, a component of chromosome axes (*Lammers et al., 1994*; *Zickler and Kleckner, 2015*), and for γH2AX, a phosphorylated form of the histone variant H2AX that is generated in response to meiotic DNA double-strand breaks (*Mahadevaiah et al., 2001*) (*Figure 1B*). In normal meiosis, SYCP3-positive axial elements begin to form during the leptotene stage of meiotic prophase I; these elongate and begin to align with homologous chromosome axes to form the tripartite synaptonemal complex in the zygotene stage; the synaptonemal complex connects homologous chromosomes along their lengths in the pachytene stage; and then the synaptonemal complex begins to disassemble during the diplotene stage (*Figure 1B*). Double-strand break formation occurs principally during leptonema and zygonema, yielding strong γH2AX staining across nuclei, but this staining diminishes as recombination proceeds (*Figure 1B*). Recombination-

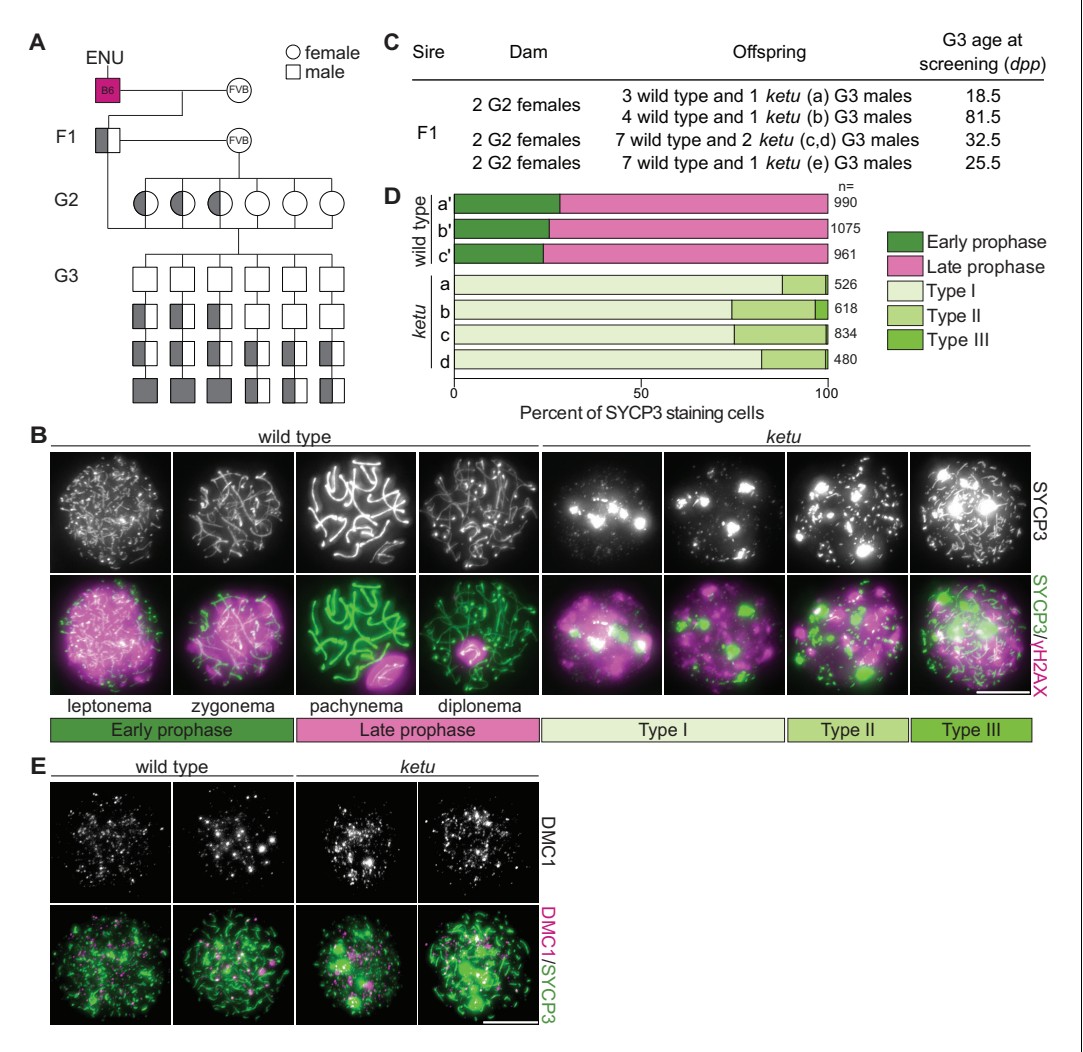

**Figure 1.** Mice from the ENU-induced mutant line *ketu* have meiotic defects. (**A**) Breeding scheme. Mutagenized males (B6) were crossed to females of a different strain (FVB) to produce founder (F1) males that were potential mutation carriers. Each F1 male was then crossed to wild-type FVB females. If an F1 male was a mutation carrier, half of his daughters (second generation, G2) should also be carriers, so the G2 daughters were crossed back to their F1 sire to generate third-generation (G3) offspring that were potentially homozygous. For a line carrying a single autosomal recessive mutation of interest, one eighth of G3 males were expected to be homozygous. Un-filled shapes represent animals that are wild-type for a mutation of interest, half-filled shapes are heterozygous carriers, and filled shapes are homozygotes. (**B**) Representative images of squashed spermatocyte preparations immunostained for SYCP3 and γH2AX. Mutant spermatocytes were classified as Types I, II, or III on the basis of SYCP3 patterns. (**C**) Screen results for the *ketu* line. The F1 male was harem-bred to six G2 females, yielding 26 G3 males that displayed either a wild-type or *ketu* (mice a, b, c, d, e) phenotype. (**D**) Distribution of SYCP3-staining patterns in four G3 *ketu* mutants (a, b, c, d) and their phenotypically wild-type littermates (a', b', c'). Wild-type spermatocytes were classified as either early prophase-like (leptonema or zygonema) or late prophase-like (pachynema or diplonema). Spermatocytes from mutant mice were categorized as described in panel B. The number of SYCP3-positive spermatocytes counted from each animal (n) is indicated and raw data are provided in *Figure 1—source data 1*. (**E**) Representative images of squashed spermatocyte preparations immunostained for DMC1 and SYCP3. Scale bars in panels B and E represent 10 μm.
DOI: https://doi.org/10.7554/eLife.30919.002

The following source data is available for figure 1:

**Source data 1.** Number of SYCP3-staining cells plotted in *Figure 1D*.
DOI: https://doi.org/10.7554/eLife.30919.003

independent γH2AX also appears in the sex body, a heterochromatic domain that encompasses the X and Y chromosomes and that is particularly evident in pachytene and diplotene cells (*Figure 1B*).

In an F1 founder line we named *ketu*, 5 of 26 G3 males screened (*Figure 1C*) contained SYCP3-positive spermatocytes displaying extreme meiotic defects, with no cells resembling normal meiotic prophase I stages (*Figure 1B,D*). For quantification, we divided mutant spermatocytes into classes on the basis of SYCP3 patterns (*Figure 1B*). Type I cells displayed few or none of the normal dots or lines of SYCP3 staining typical of early stages of axial element formation in leptonema; this was the most abundant class, accounting for 74% to 88% of cells (*Figure 1D*). Type II cells displayed prominent dots or short lines of SYCP3; these accounted for 12% to 25% of cells (*Figure 1D*). Type III cells had numerous longer lines of SYCP3, consistent with more advanced axial element formation; these were rare in the younger animals screened (<0.6%) but accumulated to slightly higher levels (3%) in the older G3 animal screened (*Figure 1C,D*). All three cell types had prominent aggregates of SYCP3 and pan-nuclear γH2AX staining (*Figure 1B*). The strong γH2AX staining is consistent with these cells having initiated meiotic recombination via formation of double-strand breaks by SPO11 (*Baudat et al., 2000*; *Mahadevaiah et al., 2001*). Supporting this interpretation, immunostaining demonstrated that *ketu* mutant spermatocytes formed numerous foci of the meiosis-specific strand exchange protein DMC1 (*Bishop et al., 1992*; *Tarsounas et al., 1999*) (*Figure 1E*). Collectively, these patterns are unlike those seen in mutants with typical meiotic recombination or synaptonemal complex defects, such as *Spo11*−/−, *Dmc1*−/−, or *Sycp1*−/−, in which recombination checkpoint or sex-body related defects occur (*Pittman et al., 1998*; *Yoshida et al., 1998*; *Baudat et al., 2000*; *Romanienko and Camerini-Otero, 2000*; *Barchi et al., 2005*; *de Vries et al., 2005*). These findings suggest that the *ketu* mutation causes a distinct defect in spermatogenesis from that seen with absence of factors involved directly in meiotic chromosome dynamics.

## *ketu* maps to a missense mutation in the *Ythdc2* gene

Because mutagenesis was carried out on B6 males, ENU-induced mutations should be linked to B6 variants for DNA sequences that differ between the B6 and FVB strains. Moreover, all *ketu*-homozygous G3 males should be homozygous for at least some of the same linked B6 variants (*Caspary, 2010*; *Horner and Caspary, 2011*). We therefore roughly mapped the *ketu* mutation by hybridizing genomic DNA from five G3 mutants to mouse SNP genotyping arrays and searching for genomic regions where all five mice shared homozygosity for B6 SNPs (*Caspary, 2010*; *Jain et al., 2017*). This yielded a 30.59-Mbp interval on Chromosome 18, flanked by heterozygous SNPs *rs4138020* (Chr18:22594209) and *gnf18.051.412* (Chr18:53102987) (*Figure 2A*). Whole-exome sequencing of DNA from mutants then revealed that this interval contained a single un-annotated DNA sequence variant located in the *Ythdc2* coding sequence (*Figure 2B,C*).

This variant is an A to G nucleotide transition at position Chr18:44840277, resulting in a missense mutation in predicted exon 6 (*Figure 2B,C*). The mutation changes codon 327 (CAT, histidine) to CGT (arginine), altering a highly conserved residue adjacent to the DEVH box (described in more detail below).

*Ythdc2* mRNA is expressed in adult testes as well as widely in other adult and embryonic tissues (*Figure 2B,D*), thus placing *Ythdc2* expression at an appropriate time to contribute to spermatogenesis. While this work was in progress, YTHDC2 protein was reported to interact in vivo with the meiosis-specific MEIOC protein, which is itself required for meiosis (*Abby et al., 2016*; *Soh et al., 2017*). Furthermore, a CRISPR/Cas9-induced frameshift mutation in exon 2 of *Ythdc2* (*Figure 2B,C*) failed to complement the *ketu* mutation (see below). We conclude that this ENU-induced point mutation disrupts *Ythdc2* function and is the cause of the *ketu* mutant phenotype in males.

## *Ythdc2*<sup>*ketu*</sup> causes male and female sterility from gametogenic failure

*Ythdc2*<sup>*ketu/+*</sup> heterozygotes had normal fertility and transmitted the mutation in a Mendelian ratio (30.8% *Ythdc2*<sup>+/+</sup>, 46.6% *Ythdc2*<sup>*ketu/+*</sup>, and 22.6% *Ythdc2*<sup>*ketu/ketu*</sup> from heterozygote × heterozygote crosses; n = 305 mice; p=0.28, Fisher's exact test). No obvious somatic defects were observed by gross histopathological analysis of major organs and tissues of adult *Ythdc2*<sup>*ketu/ketu*</sup> mice (see Materials and methods). However, *Ythdc2*<sup>*ketu/ketu*</sup> homozygous males were sterile: none of the three animals tested sired progeny when bred with wild-type females for 8 weeks. Mutant males showed a 76.4% reduction in testes-to-body-weight ratio compared to littermates (mean ratios were 0.14% for

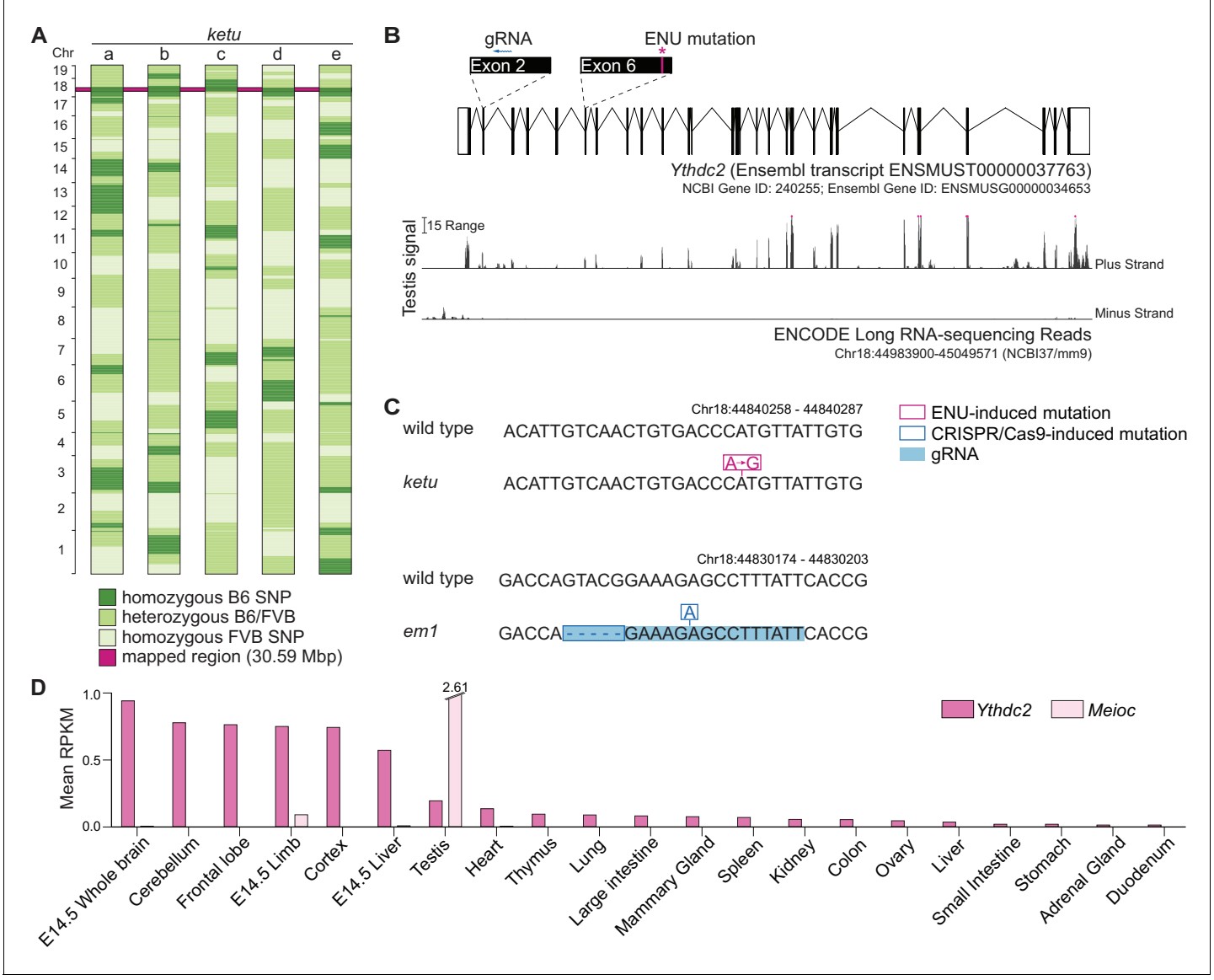

**Figure 2.** *ketu* mice harbor a point mutation in *Ythdc2*. (A) SNP genotypes of five G3 *ketu* mutants (a, b, c, d, e; from *Figure 1C*) obtained using the Illumina Medium Density Linkage Panel. The single 30.59-Mbp region of B6 SNP homozygosity that is shared between mutants is highlighted in pink. (B) Top: Schematic of *Ythdc2* (as predicted by Ensembl release 89) showing the locations of the ENU-induced lesion and the gRNA used for CRISPR/Cas9-targeting. Bottom: The density of ENCODE long RNA-sequencing reads (release 3) from adult testis within a window spanning from 3500 bp upstream to 200 bp downstream of *Ythdc2*. The vertical viewing range is 0–50; read densities exceeding this range are overlined in pink. (C) The *ketu* and CRISPR/Cas9-induced (*em1*) alleles of *Ythdc2*. (D) *Ythdc2* and *Meioc* expression level estimate (mean reads per kilobase per million mapped reads (RPKM) values provided by ENCODE; included in *Figure 2—source data 1*) in adult and embryonic tissues.

DOI: https://doi.org/10.7554/eLife.30919.004

The following source data is available for figure 2:

**Source data 1.** *Ythdc2* and *Meioc* RPKM values plotted in *Figure 2D*.
DOI: https://doi.org/10.7554/eLife.30919.005

*Ythdc2^{ketu/ketu}* and 0.58% for wild-type and heterozygous animals; p<0.01, one-sided t-test; *Figure 3A*).

In histological sections of adult testes, seminiferous tubules from *Ythdc2^{ketu/ketu}* males were greatly reduced in diameter and contained only Sertoli cells and early spermatogenic cells, with no post-meiotic germ cells (*Figure 3B*). To elucidate the timing of spermatogenic failure, we examined juveniles at 8, 10, and 14 days *post partum* (*dpp*). Meiosis first begins in male mice during the

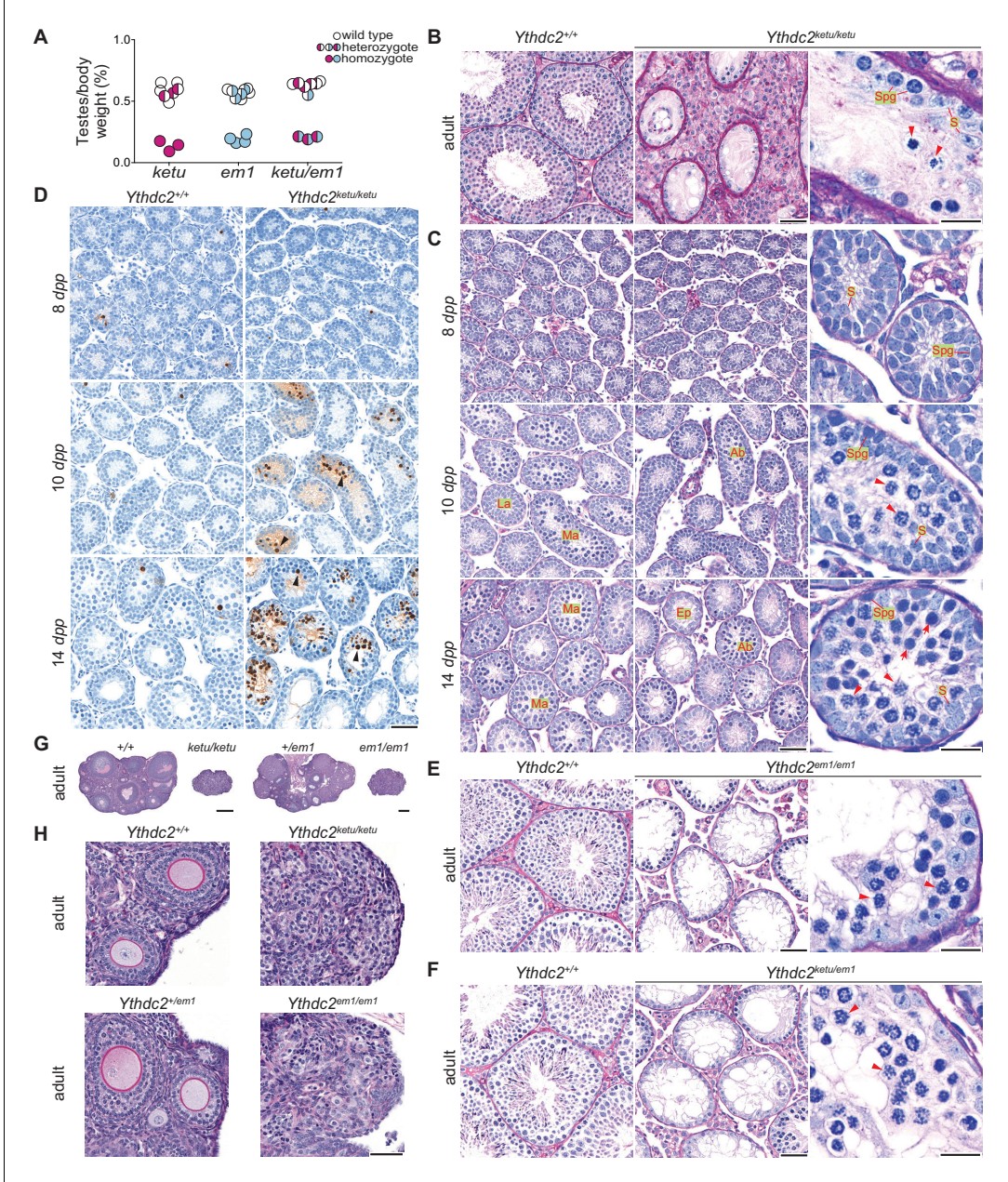

**Figure 3.** *ketu* and *em1* alleles of *Ythdc2* lead to gametogenic failure and fail to complement each other. (**A**) The ratios of testes weight to body weight for 6- to 34-week-old mice (*Figure 3—source data 1*). (**B**) PAS-stained sections of Bouin's-fixed testes from an 8-month-old *Ythdc2$^{ketu/ketu}$* male and a wild-type littermate. (**C**) PFA-fixed, PAS-stained testis sections. In wild type, examples are indicated of a less advanced ('La') tubule harboring cells with spermatogonia-like morphology and Sertoli cells, and more advanced ('Ma') tubules harboring cells with morphological characteristics of pre-leptonema and meiotic prophase stages. In the mutant, examples are indicated of abnormal ('Ab') tubules containing cells with condensed and individualized chromosomes (arrowheads), and an emptier-looking ('Ep') tubule harboring cells with spermatogonia-like morphology and Sertoli cells. (**D**) TUNEL assay on testis sections. Black arrowheads point to TUNEL-positive cells (stained dark brown). (**E** and **F**) PFA-fixed, PAS-stained testis sections from an 8-week-old *Ythdc2$^{em1/em1}$* male, a 7-week-old *Ythdc2$^{ketu/em1}$* male, and their wild-type littermates. (**G** and **H**) PFA-fixed, PAS-stained ovary sections from a 6-week-old *Ythdc2$^{ketu/ketu}$* female and a wild-type littermate, and a 9-week-old *Ythdc2$^{em1/em1}$* female and a heterozygous littermate. In the higher magnification images of panels B, C, E, and F, arrowheads indicate cells with condensed and individualized chromosomes, arrows indicate cells with morphological characteristics of pre-leptonema and leptonema, and S and Spg indicate Sertoli cells and cells with morphological characteristics of spermatogonia, respectively. In panels B, C, E, and F, the scale bars represent 50 µm and 20 µm in the lower and higher magnification images, respectively. In panels D and H, the scale bars represent 50 µm and in panel G, they represent 300 µm.
DOI: https://doi.org/10.7554/eLife.30919.006

The following source data and figure supplements are available for figure 3:

*Figure 3 continued on next page*

*Figure 3 continued*

**Source data 1.** Testes and body weights plotted in *Figure 3A*.
DOI: https://doi.org/10.7554/eLife.30919.009
**Figure supplement 1.** Quantification of TUNEL assay.
DOI: https://doi.org/10.7554/eLife.30919.007
**Figure supplement 1—source data 1.** Quantification of TUNEL assay plotted in panel A.
DOI: https://doi.org/10.7554/eLife.30919.008

second week after birth, with a population of germ cells proliferating mitotically and then entering meiosis in a semi-synchronous wave (*Bellvé et al., 1977*; *Griswold, 2016*).

At 8 *dpp*, most wild-type tubules contained only spermatogonia and Sertoli cells, as expected, and no defects were apparent in *Ythdc2*$^{ketu/ketu}$ mutants compared to normal littermate controls (*Figure 3C*). In wild type at 10 *dpp*, testes displayed a less advanced subset of tubules containing spermatogonia and Sertoli cells ('La' in *Figure 3C*), alongside more advanced tubules containing germ cells with morphological characteristics of pre-leptonema and leptonema ('Ma' in *Figure 3C*). At the same age, testis sections from *Ythdc2*$^{ketu/ketu}$ mice also had a mix of tubules at slightly different stages, but a few tubules (labeled 'Ab' in *Figure 3C*) contained germ cells with abnormal morphology in which the chromosomes were condensed and individualized (arrowheads in *Figure 3C*), reminiscent of metaphase rather than prophase. At this age in normal animals, no spermatocytes are expected to have reached even zygonema, let alone to have exited prophase (*Bellvé et al., 1977*).

By 14 *dpp*, essentially all tubules in wild type had germ cells in meiotic prophase, but *Ythdc2*$^{ketu/ketu}$ mutants displayed a mix of tubule types: some tubules ('Ab' in *Figure 3C*) had normal looking germ cells along with cells with metaphase-like chromosomes, and often contained cells with highly compacted, presumably apoptotic nuclei as well; and some had only a single layer of cells (spermatogonia plus Sertoli cells) along the tubule perimeter ('Ep' in *Figure 3C,D*). The cells with metaphase-like chromosomes were sometimes alongside cells with a morphology resembling pre-leptotene/leptotene spermatocytes (arrowheads and arrows in *Figure 3C*, respectively). Adult testes also contained cells with abnormally condensed metaphase-like chromosomes, as well as nearly empty tubules (*Figure 3B*).

We interpret the less populated tubules in juveniles and adults as those in which apoptosis has already eliminated aberrant cells. Indeed, TUNEL staining demonstrated a substantially higher incidence of apoptosis at 14 and 15 *dpp* in the *Ythdc2*$^{ketu/ketu}$ mutant compared to wild type (*Figure 3D* and *Figure 3—figure supplement 1*). Elevated apoptosis was also observed at 10 *dpp*, but not in every littermate pair of this age, presumably reflecting animal-to-animal differences in developmental timing (*Figure 3D* and *Figure 3—figure supplement 1*; see also RNA-seq section below). No increase in apoptosis was observed for *Ythdc2*$^{ketu/ketu}$ mutants at 8 *dpp* (*Figure 3D* and *Figure 3—figure supplement 1*). Apoptosis is thus contemporaneous or slightly later than the appearance of morphologically abnormal germ cells.

We conclude that *Ythdc2*$^{ketu/ketu}$ spermatogonia are able to proliferate mitotically but then transition to an aberrant state in which chromosomes condense prematurely and apoptosis is triggered. This cell death accounts for the hypogonadism, absence of postmeiotic cells in adults, and sterility.

To verify that the *Ythdc2* point mutation in the *ketu* line is causative for the spermatogenesis defect, we used CRISPR/Cas9 and a guide RNA targeted to exon 2 to generate an endonuclease-mediated (*em*) allele containing a 5 bp deletion plus 1 bp insertion, resulting in a frameshift (*Figure 2B,C*) (*Ythdc2*$^{em1Sky}$, hereafter *Ythdc2*$^{em1}$). *Ythdc2*$^{em1/+}$ heterozygotes transmitted the mutation in a Mendelian ratio (30.7% *Ythdc2*$^{+/+}$, 48.5% *Ythdc2*$^{em1/+}$, and 20.8% *Ythdc2*$^{em1/em1}$; n = 101 mice; p=0.62, Fisher's exact test). We expected that this mutation near the 5' end of the gene would be a null or cause severe loss of function. Indeed, *Ythdc2*$^{em1/em1}$ homozygotes displayed hypogonadism (0.19% mean testes-to-body-weight ratio for *Ythdc2*$^{em1/em1}$ and 0.57% for wild-type and heterozygous animals; 66.8% reduction; p<0.01, one-sided t-test) and altered testis histology indistinguishable from *Ythdc2*$^{ketu/ketu}$ homozygotes, including the appearance of cells with abnormally condensed chromosomes (*Figure 3A,E*). Moreover, *Ythdc2*$^{em1/em1}$ males were sterile (none of the three animals tested sired progeny when bred with wild-type females for 8–9 weeks). *Ythdc2*$^{ketu/em1}$ compound heterozygotes were equally defective (0.20% mean testes-to-body-weight ratio for *Ythdc2*$^{ketu/em1}$ and 0.63% for wild-type and single-heterozygous animals; 67.5% reduction; p<0.01, one-sided t-test) (*Figure 3A,F*) and sterile (two animals tested did not sire progeny when bred with wild-type females for 9 weeks). Thus, *ketu* is allelic to *Ythdc2*$^{em1}$.

$Ythdc2^{ketu/ketu}$ females were also sterile: no pregnancies were observed from crosses of 5 homozygous mutant females to wild-type males (bred for 6–19 weeks). Ovaries from $Ythdc2^{ketu/ketu}$ females were dramatically smaller compared to wild-type littermates, and no developing follicles were visible in adults (*Figure 3G,H*). Again, $Ythdc2^{em1/em1}$ homozygotes displayed the same phenotype as $Ythdc2^{ketu/ketu}$ females (*Figure 3G,H*) and no pregnancies were observed from crosses of 3 homozygous mutant females to wild-type males (bred for 9–21 weeks). These findings reveal an essential function for YTHDC2 in both male and female gametogenesis.

## Precocious meiotic progression in $Ythdc2^{ketu/ketu}$ spermatocytes

YTHDC2 and MEIOC coimmunoprecipitate from testis extracts and they bind an overlapping set of transcripts as assessed by RNA immunoprecipitation, suggesting that the two proteins function together in germ cells (*Abby et al., 2016*; *Soh et al., 2017*). This hypothesis predicts similar phenotypes for mutants defective for either gene. In mice homozygous for a targeted *Meioc* mutation, male and female germ cells enter meiosis at the correct developmental stage (i.e., in juvenile males around 10 *dpp*, and in fetal ovary around embryonic day 14.5), but show substantial meiotic defects including rapid progression to a metaphase-like state with condensed univalent chromosomes and monopolar spindles (*Abby et al., 2016*; *Soh et al., 2017*). Our initial findings from the screen showed comparable defects in *Ythdc2* mutants (*Figures 1B,D* and *3B,C,E*), so we evaluated this phenotypic similarity more closely.

To evaluate the molecular characteristics of cells containing prematurely condensed chromosomes, we immunostained 14- and 15-*dpp* testis sections to detect histone H3 phosphorylation on serine 10 (pH3), which appears at high levels on metaphase chromatin, and α-tubulin, a spindle component. In wild type, we observed groups of mitotic spermatogonia in which strong pH3 staining coated condensed chromosomes that were aligned on a metaphase plate with a bipolar spindle and pericentromeric heterochromatin (DAPI-bright regions) clustered near the middle of the metaphase plate (*Figure 4A,B,D,E* and *Figure 4—figure supplement 1A*). No spermatocytes with pH3-positive, condensed chromosomes were observed, as expected because wild-type germ cells do not reach the first meiotic division until ~18 *dpp* (*Bellvé et al., 1977*).

In $Ythdc2^{ketu/ketu}$ mutants in contrast, cells with condensed, strongly pH3-positive chromosomes were ~6 fold more abundant (*Figure 4A,C,D* and *Figure 4—figure supplement 1A*, p<0.01 comparing percent of strongly pH3-positive cells with DAPI staining indicative of condensed chromatin in juvenile wild type and mutant animals (t-test)). More importantly, unlike in wild type, most of these cells were spermatocytes as judged by their more lumenal position within the tubules (*Figure 4A,C* and *Figure 4—figure supplement 1A*) and the presence of SYCP3 at centromeres on chromosome spreads (*Figure 4F,G*). Moreover, in most of these spermatocytes the mass of condensed chromosomes had microtubules forming a single visible aster (*Figure 4C,E*). These apparently monopolar spindles contrasted starkly with those in wild-type meiosis during metaphase I (visualized in adult testes), which showed the expected bipolar α-tubulin-containing spindles and well-aligned chromosomes at mid-spindle, with pericentromeric heterochromatin at the outer edges of the metaphase plate and oriented toward the poles (*Figure 4—figure supplement 1B*).

Both in wild type and in $Ythdc2^{ketu/ketu}$ mutants, more weakly pH3-positive cells (likely spermatogonia) were also observed near tubule peripheries, with discrete pH3 foci that colocalized with the DAPI-bright pericentromeric heterochromatin (arrows in *Figure 4—figure supplement 1C*). This pattern has been reported previously (*Hendzel et al., 1997*; *Kimmins et al., 2007*; *Song et al., 2011*), and spermatogonia of this type were not obviously aberrant in the $Ythdc2^{ketu/ketu}$ mutants.

These results are consistent with the hypothesis that spermatogonial divisions occur relatively normally in *Ythdc2* mutants, but entry into and progression through meiosis are defective. To assess DNA replication in germ cells, 15-*dpp* $Ythdc2^{ketu/ketu}$ and wild-type or heterozygous littermate control animals were subjected to a 2 hr pulse label with bromodeoxyuridine (BrdU) in vivo. As expected, the mutant and controls had similar numbers of BrdU⁺ cells, inferred to be replicating spermatogonia and cells that have undergone pre-meiotic replication (*Figure 5A–C* and *Figure 5—figure supplement 1A*, p=0.54 in t-test comparing percent of BrdU⁺ cells in wild type/heterozygous and mutant). The $Ythdc2^{ketu/ketu}$ mutant had fewer SYCP3⁺ cells and those present tended to have lower SYCP3 staining intensity, presumably because of a combination of spermatocyte apoptosis (*Figure 3D*) and diminished *Sycp3* expression (see below). Importantly, however, BrdU⁺ SYCP3⁺ double-positive cells were observed in the mutant at a frequency comparable to the controls (*Figure 5C*, p=0.03 comparing percent of SYCP3⁺ cells in wild type/heterozygous and mutant, and

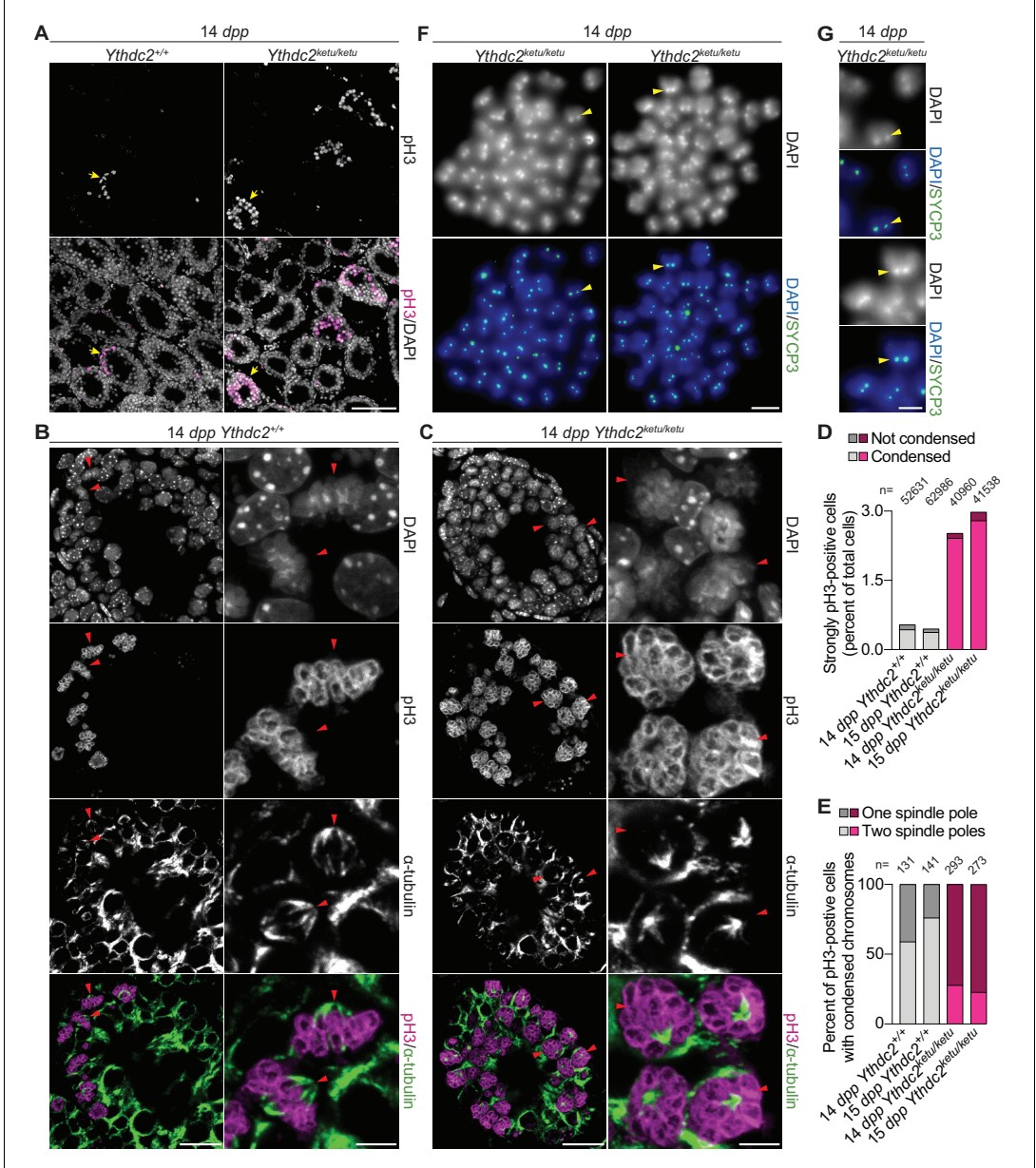

**Figure 4.** *Ythdc2*^*ketu/ketu* spermatocytes show precocious meiotic progression. (**A**) pH3 immunofluorescence on testis sections from 14-*dpp Ythdc2*^*ketu/ketu* and wild-type littermates. Yellow arrows indicate seminiferous tubules with pH3-positive cells. (**B** and **C**) α-tubulin immunofluorescence on seminiferous tubules with pH3-positive cells (tubules indicated by yellow arrows in panel A). Red arrowheads point to pH3-positive cells with DAPI-staining indicative of condensed chromatin. In each of the panels, a lower (left) and higher (right) magnification view is shown. (**D**) Quantification of pH3 immunofluorescence on testis sections from 14-*dpp*-old and 15-*dpp*-old *Ythdc2*^*ketu/ketu* and wild-type littermates. Cells with strong pH3 staining across chromatin were counted as pH3-positive and fractions of pH3-positive cells where chromatin appeared condensed (based on DAPI-staining) are indicated. The total number of testis cells counted from each animal is indicated (n). (**E**) Quantification of spindle pole number based on α-tubulin immunofluorescence on testis sections from 14-*dpp*-old and 15-*dpp*-old *Ythdc2*^*ketu/ketu* and wild-type littermates. Cells with strong pH3 staining across chromatin as well as DAPI-staining indicative of condensed chromatin were scored for the presence of one or two spindle poles. Immunostained 4 μm testis sections were imaged using three optical sections (0.6 μm step size) along the z axis. Spindle poles that fell outside the focal planes or lay outside the tissue sectioning plane would be missed. Cells were not scored if they had no visible spindle poles or if the number of spindle poles could not be discerned because of strong intercellular α-tubulin immunofluorescence. Presumably, every spindle in wild type has two poles, so the frequency of spindles with only one detectable pole in the controls provides the empirical detection failure rate. The number of cells scored from each animal is indicated (n). (**F**) SYCP3 immunofluorescence on chromosome spreads of metaphase-like cells from a 14-*dpp Ythdc2*^*ketu/ketu* animal. 37–40 SYCP3 foci doublets (yellow arrowheads) are visible in each spread. (**G**) Higher magnification view of SYCP3 foci doublets (corresponding to the yellow arrowheads in panel F), showing the colocalization of SYCP3 signal with the DAPI-bright pericentric regions. In panels A, F and G, the scale bars represent 100 μm,

*Figure 4 continued on next page*

Figure 4 continued

10 µm, and 5 µm, respectively. In panels B and C, the scale bars represent 20 µm and 5 µm in the lower (left) and higher (right) magnification images, respectively. Raw data for panels D and E are provided in *Figure 4—source data 1* and *Figure 4—source data 2*, respectively.

DOI: https://doi.org/10.7554/eLife.30919.010

The following source data and figure supplement are available for figure 4:

**Source data 1.** Quantification of pH3 immunofluorescence plotted in *Figure 4D*.
DOI: https://doi.org/10.7554/eLife.30919.012
**Source data 2.** Quantification of spindle poles plotted in *Figure 4E*.
DOI: https://doi.org/10.7554/eLife.30919.013
**Figure supplement 1.** α-tubulin and pH3 immunofluorescence in *Ythdc2$^{ketu}$* mutants.
DOI: https://doi.org/10.7554/eLife.30919.011

p=0.53 comparing percent of double-positive cells (t-test)). The presence of these presumptive preleptotene spermatocytes suggests that pre-meiotic DNA replication occurs in a timely manner in the *Ythdc2$^{ketu/ketu}$* mutant, similar to *Meioc$^{-/-}$* (*Abby et al., 2016*; *Soh et al., 2017*), although modest defects in timing or efficiency cannot be ruled out.

In normal spermatogenesis, cyclin A2 (CCNA2) is expressed during mitotic cell cycles in spermatogonia, but is downregulated at or before meiotic entry and is undetectable in SYCP3-positive spermatocytes (*Ravnik and Wolgemuth, 1999*). *Meioc$^{-/-}$* spermatocytes fail to properly extinguish CCNA2 expression, suggesting that inappropriate retention of the spermatogonial (mitotic) cell cycle machinery contributes to precocious metaphase and other meiotic defects (*Soh et al., 2017*). When testis sections from 12-, 14-, and 15-*dpp* wild-type animals were immunostained for SYCP3 and CCNA2, the expected mutually exclusive localization pattern was observed: CCNA2$^+$ spermatogonia were located at tubule peripheries while SYCP3$^+$ spermatocytes occupied more lumenal positions (*Figure 5D*, left), and double-positive cells were rare (*Figure 5E,F* and *Figure 5—figure supplement 1B*). In testis sections from *Ythdc2$^{ketu/ketu}$* animals of the same ages, however, a substantial fraction of SYCP3$^+$ spermatocytes also contained nuclear CCNA2 (*Figure 5D*, right, *Figure 5E,F*, and *Figure 5—figure supplement 1B*). Indeed, the *Ythdc2* mutant showed a net increase in the number of CCNA2$^+$ cells, and much of this increase can be attributed to cells that initiated a meiotic program (i.e., were SYCP3$^+$), but a small increase in the spermatogonial pool cannot be ruled out (*Figure 5F*, p<0.01 comparing wild type and mutant animals for percent of SYCP3$^+$ cells, or CCNA2$^+$ cells, or double-positive cells (t-test)).

Collectively, these data lead us to conclude that *Ythdc2*-defective germ cells make an abortive attempt to enter meiosis but fail to fully extinguish the spermatogonial program, progress precociously to a metaphase-like state, and undergo apoptosis. The *Ythdc2* and *Meioc* mutant phenotypes are highly concordant in this regard (*Abby et al., 2016*; *Soh et al., 2017*), supporting the hypothesis that YTHDC2 and MEIOC function together to regulate germ cell development around the time of meiotic entry.

### *Ythdc2* mutant germ cells are defective at transitioning to the meiotic RNA expression program

To test whether the deregulation of meiotic progression in the absence of YTHDC2 was accompanied by alterations in the transcriptome, we performed RNA sequencing (RNA-seq) on whole testes from *Ythdc2$^{em1/em1}$* and wild-type or heterozygous littermate controls. We analyzed steady-state RNA levels at 8 and 9 *dpp*, at or slightly before initiation of meiosis, and at 10 *dpp*, by which time aberrant metaphases were observed in the mutant (*Figure 3*). At these ages for the experimental cohort, mutant testes did not appear to contain elevated numbers of TUNEL-positive cells compared to control littermates (*Figure 6—figure supplement 1A*), indicating that programmed cell death had not yet significantly altered the relative sizes of germ cell subpopulations in these samples.

We first examined whether genes were significantly altered in the mutant for each day in the time course. At a modestly stringent statistical cutoff (adjusted p-value<0.05), only small numbers of genes (a dozen or fewer) scored as significantly up- or downregulated in the mutants at 8 and 9 *dpp* (*Figure 6A*, *Figure 6—figure supplement 1B* and *Supplementary file 1*). At 10 *dpp*, a similarly small number of genes were upregulated, but a larger group (69 genes) was significantly downregulated (*Figure 6A*, *Figure 6—figure supplement 1B* and *Supplementary file 1*). We performed

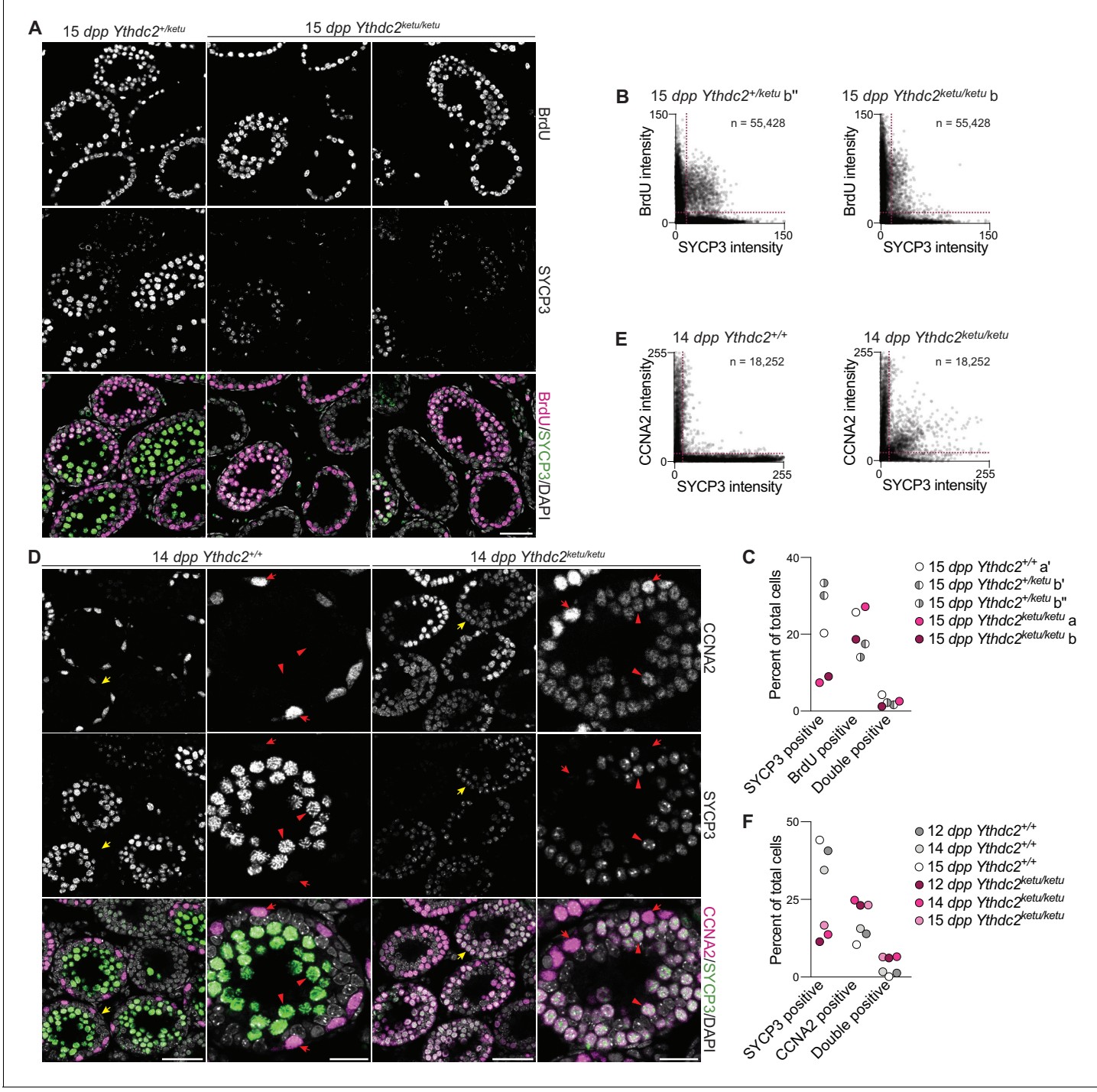

**Figure 5.** Mixed mitotic and meiotic characteristics of *Ythdc2*[ketu/ketu] spermatocytes. (**A**) BrdU and SYCP3 immunofluorescence on testis sections from 15-*dpp Ythdc2*[ketu/ketu] and heterozygous littermates. Testes were fixed 2 hr after BrdU administration. (**B**) Scatter-plots showing BrdU and SYCP3 immunofluorescence intensity of testis cells from 15-*dpp Ythdc2*[ketu/ketu] (b) and heterozygous (b″) littermates. Immunostained testis sections were imaged and average fluorescence intensities of individual cells were measured as 8-bit intensity values (0–255 AU). The number of cells scored from each animal is indicated (n) and dotted lines denote intensity thresholds applied for scoring cells as positively stained. (**C**) Quantification of BrdU and SYCP3 immunofluorescence on testis sections from 15-dpp-old *Ythdc2*[ketu/ketu] (a, b) and wild-type or heterozygous (a′, b′, b″) littermates (same mice analyzed in panel B and *Figure 5—figure supplement 1A*). Fluorescence intensities were measured as described for panel B and a minimum intensity threshold of 15 AU was applied for scoring cells as positive for staining. (**D**) CCNA2 and SYCP3 immunofluorescence on testis sections from 14-*dpp Ythdc2*[ketu/ketu] and wild-type littermates. Yellow arrows in the lower magnification views (first and third columns) point to tubules of interest, which are shown at a higher magnification in the second and fourth columns. In the higher magnification views, red arrows point to cells with spermatogonia-like

*Figure 5 continued on next page*

*Figure 5 continued*

morphology and red arrowheads indicate SYCP3-positive spermatocytes. (E) Scatter-plots showing CCNA2 and SYCP3 immunofluorescence intensity of testis cells from 14-*dpp Ythdc2^ketu/ketu* and wild-type littermates. Fluorescence intensities were measured as described for panel B and the number of cells scored from each animal is indicated (n). (F) Quantification of CCNA2 and SYCP3 immunofluorescence on testis sections from *Ythdc2^ketu/ketu** and wild-type littermates of the indicated ages (same mice analyzed in panel E and *Figure 5—figure supplement 1B*). Fluorescence intensities were measured as described for panel B and a minimum intensity threshold of 20 AU was applied for scoring cells as positive for staining. In panel A, the scale bar represents 50 μm. In panel D, the scale bars represent 50 μm and 20 μm in the lower and higher magnification images, respectively. Raw data for panels B, C, E and F are provided in *Figure 5—source data 1*, *Figure 5—source data 2*, *Figure 5—source data 3* and *Figure 5—source data 4*, respectively.

DOI: https://doi.org/10.7554/eLife.30919.014

The following source data and figure supplement are available for figure 5:

**Source data 1.** Quantification of BrdU and SYCP3 immunofluorescence plotted in *Figure 5B*.
DOI: https://doi.org/10.7554/eLife.30919.016
**Source data 2.** Quantification of BrdU and SYCP3 immunofluorescence plotted in *Figure 5C*.
DOI: https://doi.org/10.7554/eLife.30919.017
**Source data 3.** Quantification of CCNA2 and SYCP3 immunofluorescence plotted in *Figure 5E*.
DOI: https://doi.org/10.7554/eLife.30919.018
**Source data 4.** Quantification of CCNA2 and SYCP3 immunofluorescence plotted in *Figure 5F*.
DOI: https://doi.org/10.7554/eLife.30919.019
**Figure supplement 1.** BrdU and CCNA2 immunofluorescence on *Ythdc2^ketu^* mutants.
DOI: https://doi.org/10.7554/eLife.30919.015

hierarchical clustering to divide into similar-behaving groups all genes that scored as significantly differently expressed in the mutant during at least one period in the developmental time course (n = 113 genes), and the dendrogram obtained showed four broad clusters (*Figure 6B*, see Materials and methods). Cluster I comprised genes whose expression remained low during these time points in wild type but were generally upregulated in *Ythdc2^em1/em1^* mutants. Clusters II and III comprised genes whose expression in wild type rose sharply between 9 and 10 *dpp* but that mostly failed to increase in mutants. And Cluster IV comprised genes whose expression in wild type increased more gradually or remained high over this period but that remained consistently lower in mutants. Although the differential expression in mutants was most dramatic at 10 *dpp*, we note that a similar direction of misregulation was generally also seen at 8 and 9 *dpp*.

To better understand these gene regulation changes, we compared our differentially regulated gene set to published RNA-seq data from sorted testis cell populations (*Soumillon et al., 2013*) (*Figure 6C*). Cluster I was enriched for genes whose expression was higher in sorted spermatogonia compared to spermatocytes (p<0.0001; paired t-test), while Clusters II and III were enriched for genes with the opposite developmental pattern (p<0.01). Cluster IV genes were not significantly different between spermatogonia and spermatocytes (p=0.086; listed in *Figure 6—source data 1*). Furthermore, Clusters II and III were highly enriched for meiosis-related gene ontology (GO) terms (adjusted p-value<0.05, *Figure 6D*; Clusters I and IV were not significantly enriched for any GO terms). These results are consistent with the idea that *Ythdc2* mutants have defects in fully downregulating the spermatogonial program and establishing at least a subset of the meiotic transcriptome.

To test this hypothesis further and to more clearly delineate when RNA expression levels began to change, we examined the behavior of developmentally regulated genes more closely. For this purpose, we defined developmentally regulated genes as those whose expression either increased or decreased in wild type between 8 *dpp* and 10 *dpp* (adjusted p-value<0.1). Strikingly, when subdivided according to developmental fate in wild type, these genes showed opposing behaviors in mutants at 8 *dpp*: the genes that would be destined to increase over time in wild type tended to be expressed at lower levels in mutants at 8 *dpp*, while genes that should be destined to decrease over time tended to be overexpressed in the mutant (*Figure 6E*; p=2.6×10^{-244} comparing the two developmentally regulated groups). Similar results were obtained if we restricted the developmentally regulated gene set to include only those transcripts that were also reported to be enriched for $N^6$-methyladenosine (m$^6$A) at 8 *dpp* (*Hsu et al., 2017*), a potential binding target of the YTH domain (discussed further below) (*Figure 6E*).

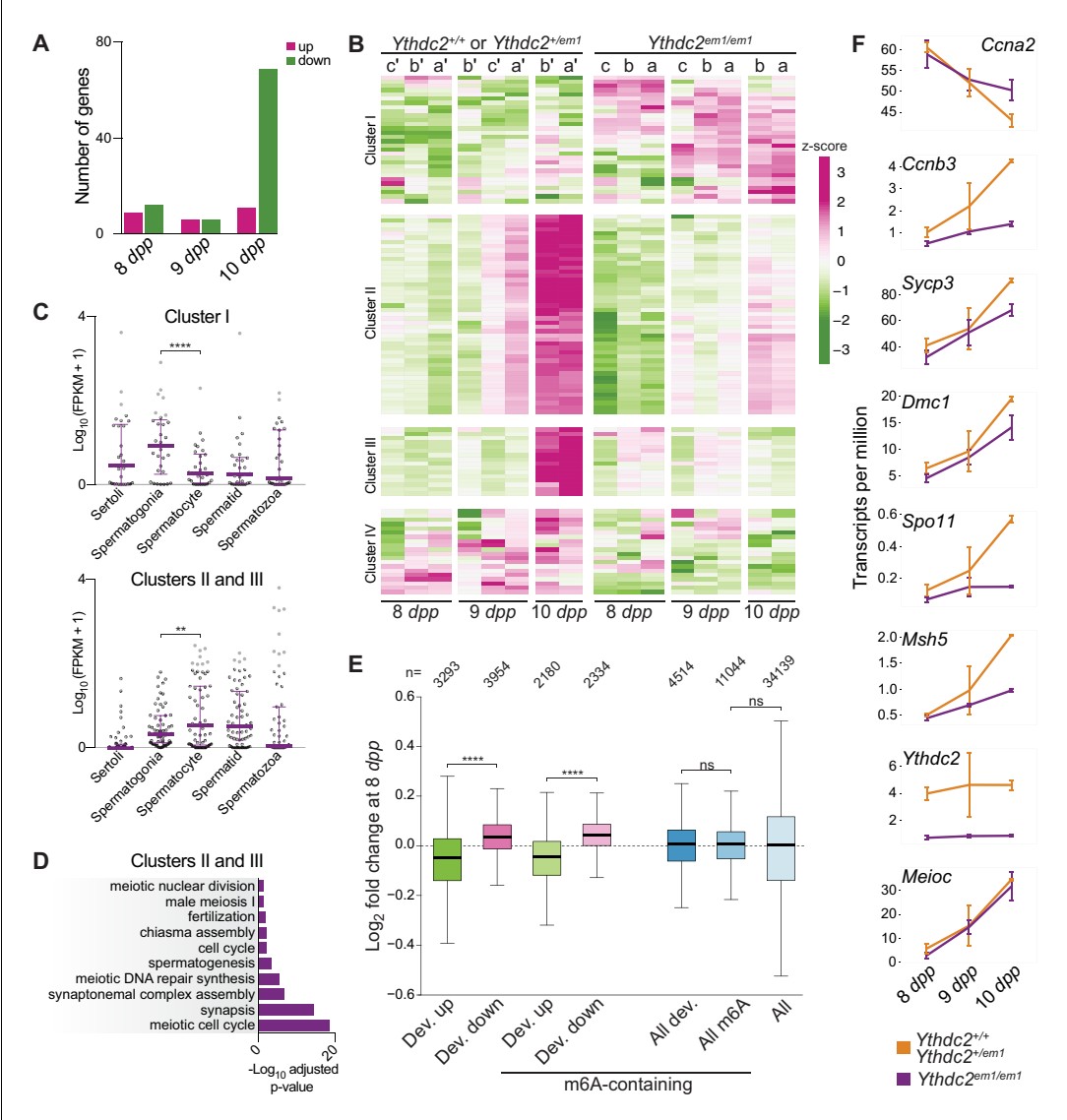

**Figure 6.** RNA-seq analysis of *Ythdc2em1* mutants during the germline transition into meiosis. (**A**) Number of differentially regulated genes (Benjamini-Hochberg-adjusted p-value<0.05 from DESeq2) in pairwise comparisons of mutants with controls at 8, 9, and 10 *dpp* (see Materials and methods for a breakdown of mice used). (**B**) Heatmap showing the expression z-scores of differentially regulated genes in individual *Ythdc2em1* mutants (a, b, c) and their wild-type or heterozygous littermates (a′, b′, c′). Genes that were differentially regulated (adjusted p-value<0.05) in mutant vs. control pairwise comparisons of mice of a specific age (8 *dpp*; 9 *dpp*; 10 *dpp*) or groups of ages (8 and 9 *dpp* combined; 9 and 10 *dpp* combined; 8, 9 and 10 *dpp* combined) are shown. Differentially regulated genes were divided into four clusters (see Materials and methods). (**C**) Expression levels of differentially regulated genes belonging to Cluster I or Clusters II and III in purified testicular cell types (*Soumillon et al., 2013*). Medians and interquartile ranges are shown. Quadruple-asterisk represents p<0.0001 and double-asterisk represents p<0.01 in paired t-tests. (**D**) Benjamini-Hochberg-adjusted p-values from DAVID (*Huang et al., 2009a*; *b*) of enriched GO terms (adjusted p-value<0.05) for differentially regulated genes belonging to Clusters II and III. (**E**) Expression fold changes of developmentally regulated genes at 8 *dpp*. Developmentally regulated genes were defined as those with increased (Dev. up) or decreased (Dev. down) expression (adjusted p-value<0.1) in a pairwise comparison between 8 *dpp*-old and 10 *dpp*-old wild-type or heterozygous animals. Developmentally regulated genes that contain m6A peaks are shown with genes grouped irrespective of their developmental fate (m6A-containing, All dev.) or as those with increased (m6A-containing, Dev. up) or decreased (m6A-containing, Dev. down) expression from 8 *dpp* to 10 *dpp*. All m6A-containing genes (All m6A) and the entire transcriptome (All) are also shown for comparison. Quadruple-asterisk represents p<0.0001 and ns (not significant) represents p>0.05 in Wilcoxon rank sum test. The number of genes in each category is indicated (n). (**F**) Mean transcript levels of indicated genes at 8, 9 and 10 *dpp*. Error bars indicate standard deviations. Expression fold changes of differentially regulated genes shown in panel B are in *Supplementary file 1*. Raw data for panels C, D and E are provided in *Figure 6—source data 1*, *Figure 6—source data 2* and *Figure 6—source data 3*, respectively.

DOI: https://doi.org/10.7554/eLife.30919.020

*Figure 6 continued on next page*

*Figure 6 continued*

The following source data and figure supplement are available for figure 6:

**Source data 1.** FPKM values of differentially expressed genes in purified testicular cells plotted in *Figure 6C*.
DOI: https://doi.org/10.7554/eLife.30919.022
**Source data 2.** p-values of significantly enriched GO terms in Clusters II and III plotted in *Figure 6D*.
DOI: https://doi.org/10.7554/eLife.30919.023
**Source data 3.** Expression fold changes of developmentally regulated genes plotted in *Figure 6E*.
DOI: https://doi.org/10.7554/eLife.30919.024
**Figure supplement 1.** RNA-seq analysis of *Ythdc2^em1* mutants.
DOI: https://doi.org/10.7554/eLife.30919.021

*Figure 6F* displays behaviors of transcripts for selected genes in the RNA-seq dataset. *Ccna2* transcript levels declined over the time points examined in wild type, but went down to a lesser degree at 10 *dpp* in mutants. Although this difference was not large enough to meet our cutoff for statistical significance ($\log_2$ fold change at 10 *dpp* = 0.23 with adjusted p-value=0.26), it is consistent with the observed persistence of CCNA2 protein expression (*Figure 5D,F*). In comparison, the transcript level of *Ccnb3*, encoding a meiotic cyclin expressed during early prophase I (*Nguyen et al., 2002*), was decreased at 10 *dpp* in mutants compared to wild type ($\log_2$ fold change at 10 *dpp* = −1.60 with adjusted p-value<0.01), as were transcripts for other canonical meiotic genes *Sycp3*, *Dmc1*, *Spo11*, and *Msh5* ($\log_2$ fold changes at 10 *dpp* = −0.42, −0.46, −1.97, −1.05, respectively; adjusted p-value<0.01 for *Sycp3*, *Spo11*, *Msh5*, and 0.06 for *Dmc1*). *Ythdc2* transcripts were decreased in mutants at all ages examined ($\log_2$ fold change at 8, 9, 10 *dpp* = −2.50, −2.52, −2.42, respectively, with adjusted p-value<0.01), most likely reflecting nonsense-mediated decay caused by the frameshift mutation. *Meioc* was unaffected during this developmental window ($\log_2$ fold change at 8, 9, 10 *dpp* = −1.02, −0.10, −0.12, respectively, with adjusted p-value=1.0).

Collectively, these findings support the conclusion that YTHDC2 is important for the developmental transition from a spermatogonial to a meiotic gene expression program and that RNA changes appear before overt morphological changes or cellular attrition through programmed cell death occur. However, we emphasize that the influence of YTHDC2 on transcript abundance (whether direct or indirect) is quantitatively modest and can vary substantially in direction between affected transcripts. Because the *Ythdc2* mutation has opposite effects on groups of transcripts with distinct developmental fates, effects of the mutation were obscured if all m^6A-enriched transcripts were pooled (*Figure 6E*, right). Decreased levels of meiotic transcripts were also observed in microarray analyses of 8-*dpp* *Meioc*^−/− testes (*Abby et al., 2016*), and a similar overall pattern of downregulated meiotic genes and/or upregulated mitotic genes was seen with microarray or RNA-seq data from *Meioc*^−/− ovaries at embryonic day 14.5 (*Abby et al., 2016*; *Soh et al., 2017*). Although technical differences between these experiments preclude a direct analysis of one-to-one correspondence, the broad strokes show congruent patterns of dysregulation of the stem-cell-to-meiosis transition in both *Meioc*^−/− and *Ythdc2*^em1/em1 mutants, lending further support to the conclusion that YTHDC2 and MEIOC function together.

## YTHDC2 localizes to the cytoplasm of prophase I spermatocytes

To determine the temporal and spatial distribution of YTHDC2 protein in the male germline, we immunostained testis sections from adult wild-type and mutant animals (*Figure 7* and *Figure 7—figure supplement 1*). YTHDC2 staining was prominent in SYCP3-positive spermatocytes in wild type, whereas little to no staining was observed in spermatocytes from *Ythdc2*^em1/em1 littermates (*Figure 7A*), validating antibody specificity.

In contrast to wild-type littermates, *Ythdc2*^ketu/ketu mutants displayed only background levels of YTHDC2 staining in SYCP3-positive cells (arrowheads in *Figure 7B*), similar to *Ythdc2*^em1/em1 mutants. The *Ythdc2*^ketu mutation may destabilize the protein, and/or YTHDC2 may be required (directly or indirectly) for its own expression as cells transition into meiosis (discussed further below).

In wild type, no YTHDC2 staining above background was observed in spermatogonia or Sertoli cells (e.g., exemplified by the stage I–III tubule in *Figure 7C*). However, given that testes from 8-*dpp* wild-type animals express *Ythdc2* mRNA (*Figure 6F*) and mutants show altered patterns of RNA expression at this age (*Figure 6E*), YTHDC2 protein may be present in late-stage differentiating

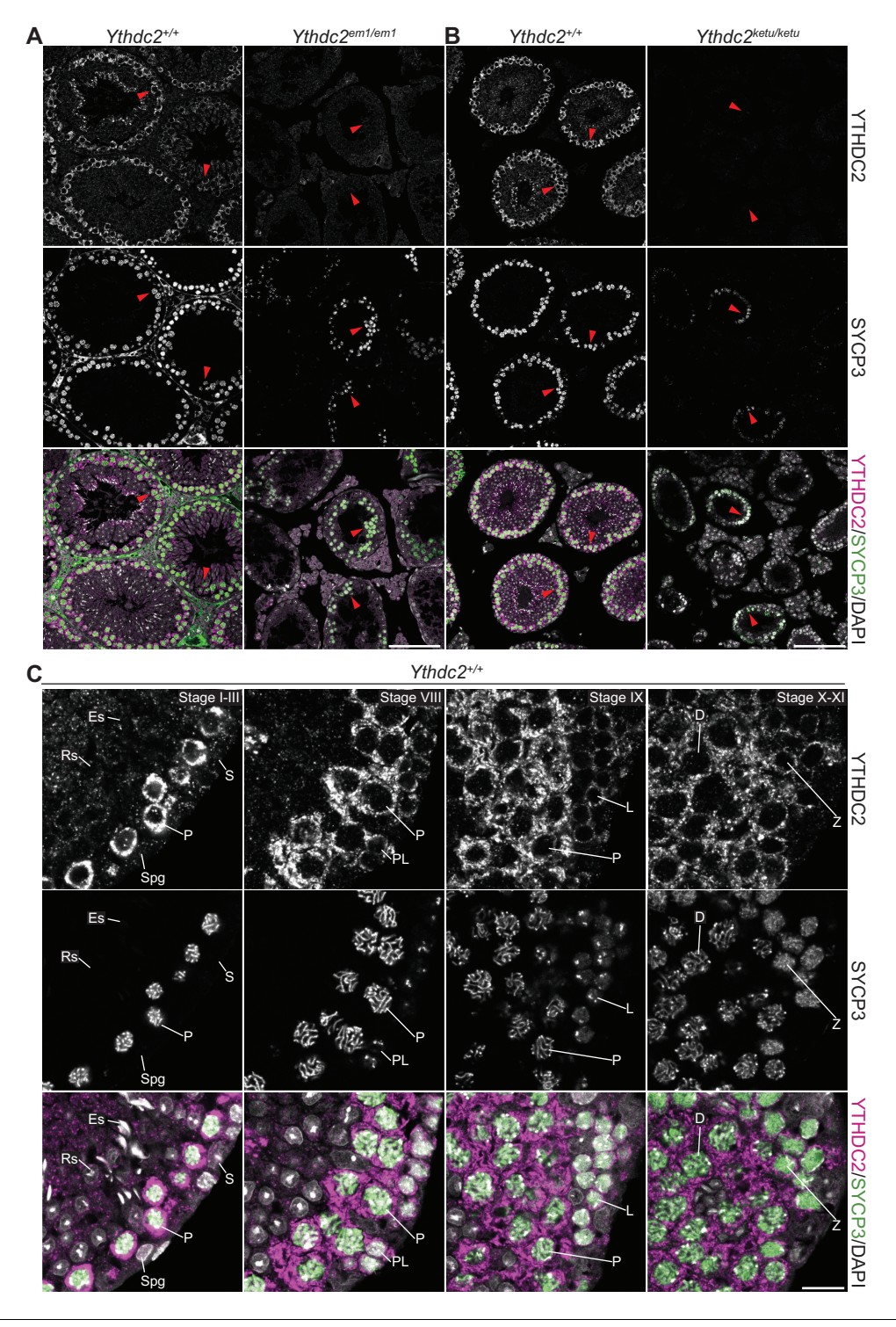

**Figure 7.** YTHDC2 localization in wild-type and mutant testes. (**A** and **B**) YTHDC2 and SYCP3 immunofluorescence on testis sections from 2-month-old *Ythdc2^{em1/em1}*, *Ythdc2^{ketu/ketu}*, and their wild-type littermates. Arrowheads indicate SYCP3-positive spermatocytes. Scale bars represent 100 μm. (**C**) YTHDC2 and SYCP3 immunofluorescence on testis sections from an adult B6 male. Approximate seminiferous epithelial cycle stages (based on SYCP3 and DAPI staining patterns) are provided. S, Sertoli cell; Spg, spermatogonia; L, leptotene spermatocyte; Z, zygotene spermatocyte; P, pachytene spermatocyte; D, diplotene spermatocyte; RS, round spermatid; ES, elongating spermatid. Scale bar represents 15 μm.

*Figure 7 continued on next page*

*Figure 7 continued*

DOI: https://doi.org/10.7554/eLife.30919.025

The following figure supplement is available for figure 7:

**Figure supplement 1.** YTHDC2 staining with an independent anti-YTHDC2 antibody in wild-type and mutant testes.

DOI: https://doi.org/10.7554/eLife.30919.026

spermatogonia at levels difficult to detect above background with these antibodies. Strong staining first became detectable in pre-leptotene spermatocytes (stage VIII tubule, *Figure 7C*), remained strong from leptonema through diplonema, then returned to background levels in postmeiotic cells (round spermatids) (*Figure 7C*). YTHDC2 appeared cytoplasmic throughout prophase I, with no detectable nuclear signal above background. These localization patterns were reproducible with an independent anti-YTHDC2 antibody (*Figure 7—figure supplement 1*). Our observations agree with and extend prior reports of cytoplasmic YTHDC2 staining in meiotic prophase I spermatocytes (*Abby et al., 2016*; *Soh et al., 2017*).

## Structure and domain architecture of YTHDC2

The predicted *Ythdc2* transcript encodes a protein of 1445 amino acids and 161 kDa (presented schematically in *Figure 8A*). Protein domains and amino acid sequence motifs characteristic of superfamily 2 DExH-box helicases are conserved in mouse YTHDC2 and its homologs throughout Eumetazoa (*Figure 8A,B*). More specifically, YTHDC2 is predicted to have the helicase core modules (DEXDc and HELICc domains, including matches to helicase conserved sequence motifs) and two C-terminal extensions (the helicase-associated 2 (HA2) and oligonucleotide binding (OB) domains) that are characteristic of the DEAH/RNA helicase A (RHA) helicase family (*Figure 8A,B* and *Supplementary file 2*) (*Fairman-Williams et al., 2010*). The *ketu* mutation (H327R) alters the amino acid located four residues N-terminal of the DEVH sequence (*Figure 8B*). This histidine is invariant across likely YTHDC2 orthologs (*Figure 8B*).

On the basis of sequence alignments, human and mouse YTHDC2 share similarity with other DEAH-box proteins DHX30, DHX9, and TDRD9 (*Figure 8C*), but YTHDC2 is decorated with several auxiliary domains not found in these other proteins, namely, an N-terminal R3H domain, an ankyrin repeat domain (ARD) inserted between the two helicase core domains and containing a pair of ankyrin repeats, and a C-terminal YTH domain (*Figure 8A* and *Figure 8—figure supplement 1A*). R3H domains are implicated in nucleic acid binding and protein oligomerization (*He et al., 2013*; *He and Yan, 2014*). The ARD may be involved in protein-protein interactions (*Li et al., 2006*).

Most characterized YTH domains bind specifically to RNA containing m⁶A (*Dominissini et al., 2012*; *Meyer et al., 2012*; *Schwartz et al., 2013*; *Wang et al., 2014*; *Xu et al., 2015*; *Patil et al., 20172018*). A crystal structure of the YTH domain of human YTHDC1 bound to 5′-GG(m⁶A)CU revealed that the methyl group in m⁶A is accommodated in a pocket composed of three aromatic/hydrophobic residues (*Xu et al., 2014*). These residues are present in the YTH domains of human and mouse YTHDC2 (triangles in *Figure 8D*). To evaluate this conservation in more detail, we examined an NMR structure of the YTH domain of human YTHDC2 (RIKEN Structural Genomics Consortium; *Figure 8E*). The YTHDC2 YTH domain adopts an open α/β fold with a core composed of six β strands surrounded by four alpha helices. This solution structure matched closely with the crystal structure of RNA-bound YTHDC1 (Cα r.m.s.d. = 2.27 Å) and, importantly, the conserved m⁶A-binding residues aligned well (*Figure 8F*). Consistent with this structural conservation, the human YTHDC2 YTH domain was reported to bind m⁶A-modified RNAs, albeit with substantially weaker affinity than YTH domains from other proteins (*Xu et al., 2015*).

The YTH domain of the *Schizosaccharomyces pombe* Mmi1 protein is an exception to the more widely found m⁶A-binding specificity. Rather than binding m⁶A, the Mmi1 YTH domain binds a specific RNA sequence called a 'determinant of selective removal' (DSR: 5′-UNAAA/C) found in transcripts of meiotic and other genes (*Harigaya et al., 2006*; *Yamashita et al., 2012*; *Chatterjee et al., 2016*). Crystal structures of the *S. pombe* Mmi1 YTH domain alone and bound to RNA with a DSR motif implicated an RNA interaction site distinct from the protein surface by which other YTH domain proteins bind m⁶A-modified RNA (*Chatterjee et al., 2016*; *Wang et al., 2016*).

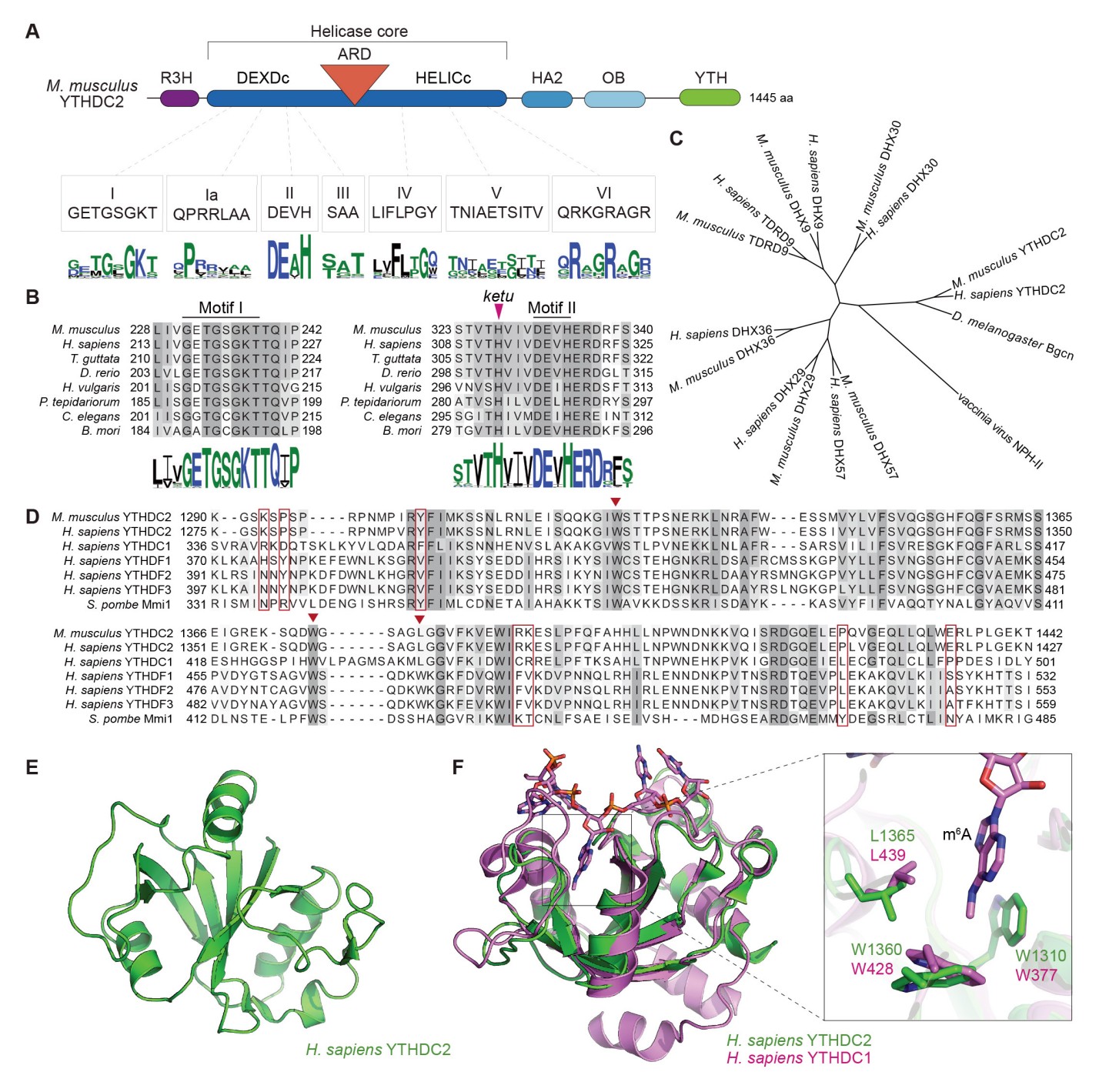

**Figure 8.** YTHDC2 domain architecture and structure of its YTH domain. (**A**) Schematic of mouse YTHDC2 domain structure (not to scale). Sequence motifs characteristic of superfamily 2 DExH-box helicases (I, Ia, II, III, IV, V, VI) within the helicase core domain are indicated, along with sequence logos from Clustal Omega alignments of 157 superfamily 2 DExH-box helicases. The height of each stack of residues reflects the degree of conservation, and the height of each amino acid symbol within each stack is proportional to the frequency of a residue at that position. Amino acids are colored according to their physico-chemical properties (hydrophilic (blue), neutral (green), and hydrophobic (black)). (**B**) Clustal Omega alignments of sequences around helicase motifs I and II from YTHDC2 proteins of the indicated species. The position of the *ketu* mutation is indicated. The residues are shaded based on percent agreement with the consensus. Sequence logos were generated from Clustal Omega alignments of YTHDC2 homologs from 201 species and are colored as in panel A. (**C**) Cladogram of Clustal Omega protein sequence alignments of mouse and human YTHDC2 paralogs. The tree was rooted using vaccinia virus NPH-II sequence and *D. melanogaster* Bgcn is included for comparison. DHX29 is an RNA helicase that promotes translation initiation of mRNAs with structured 5' untranslated regions (**Pisareva et al., 2008**). DHX57 is an uncharacterized protein of unknown

*Figure 8 continued on next page*

*Figure 8 continued*
function. DHX36 has G-quadruplex unwinding activity for DNA and RNA and is involved in antiviral responses to dsRNA (*Vaughn et al., 2005*; *Yoo et al., 2014*). DHX9 (also known as RNA helicase A and DNA helicase II) has both DNA and RNA helicase activities and multiple cellular functions, including in genome stability and in viral and cellular RNA metabolism (*Friedemann et al., 2005*; *Lin et al., 2012*). DHX30 is a poorly characterized protein required for cell viability during mouse embryonic development (*Zheng et al., 2015*). And TDRD9 forms a complex with MIWI2 involved in piRNA-directed transposon silencing in the male germline (*Shoji et al., 2009*; *Wenda et al., 2017*). (D) Clustal Omega alignment of YTH domain sequences. Inverted triangles in auburn indicate residues that make up the hydrophobic pocket that binds m6A. Residues boxed in auburn are required for Mmi1 interaction with RNA (*Wang et al., 2016*). (E) NMR structure of the YTH domain from human YTHDC2, generated by the RIKEN Structural Genomics/Proteomics Initiative (*Endo et al., 2007*) (PDB ID: 2YU6). (F) Structure of the m6A binding pocket is conserved in YTHDC2. The solution structure of YTH domain from human YTHDC2 (green) is shown superimposed on the crystal structure of the RNA-bound YTH domain from human YTHDC1 (pink; PDB ID: 4R3I) (*Xu et al., 2014*). The m6A nucleotide and the hydrophobic amino acid residues lining the binding pocket are shown at a higher magnification on the right. Protein accession numbers for sequence logos in panel A are in *Supplementary file 2*; all other protein accession numbers are in *Supplementary file 3*.

DOI: https://doi.org/10.7554/eLife.30919.027

The following figure supplement is available for figure 8:

**Figure supplement 1.** Domain architecture of YTHDC2 and related DEAH-box proteins.
DOI: https://doi.org/10.7554/eLife.30919.028

Although the human YTHDC2 structure superimposes well on that of Mmi1 (*Figure 8—figure supplement 1B*), several key residues involved in DSR binding are not conserved in mouse and human YTHDC2 (red boxes in *Figure 8D*). Specifically, RNA binding was shown to be compromised by mutation of Mmi1 Tyr-466, the side-chain hydroxyl of which forms a hydrogen bond with the $N^1$ atom of DSR nucleobase $A_4$ (*Wang et al., 2016*). This position is a proline in both human and mouse YTHDC2 (*Figure 8D* and *Figure 8—figure supplement 1B*). We infer that it is unlikely that the YTH domain of YTHDC2 utilizes the DSR interaction surface to bind RNA.

## YTHDC2 is a 3′→5′ RNA helicase

YTHDC2 has been shown to have RNA-stimulated ATPase activity (*Morohashi et al., 2011*) and was sometimes referred to as an RNA helicase (e.g.,*Tanabe et al., 2016*; *Soh et al., 2017*), but direct demonstration of helicase activity had not yet been reported at the time our studies were conducted. To test the helicase activity, we expressed and purified the recombinant mouse YTHDC2 protein from insect cells (*Figure 9A*) and performed a strand displacement assay using RNA duplexes with blunt ends, a 5′ overhang, or 3′ overhang. One strand was labeled with 6-carboxy-fluorescein (6-FAM) to follow its dissociation from the duplex substrate and a DNA trap was present to prevent re-annealing of the displaced strand. We compared substrates with 25-nt poly(A) or poly (U) overhangs to evaluate sequence specificity. Unwinding was observed for the 3′ tailed substrates but not for substrates with blunt ends or 5′ overhangs (*Figure 9B*), indicating a 3′→5′ helicase activity. ATP was required for duplex unwinding (*Figure 9C*). Under these conditions, YTHDC2 was more active on duplexes with a 3′ poly(U) overhang than a 3′ poly(A) overhang (*Figure 9B*), and at a set enzyme concentration, it unwound the 3′ poly(U) overhang substrate more efficiently (*Figure 9D,E*).

Introduction of the *ketu* mutation in YTHDC2 led to protein insolubilty when expressed in insect cells (*Figure 9—figure supplement 1A*). A small fraction of soluble protein could be retrieved using a construct with an N-terminal maltose-binding protein (MBP) tag. Purified MBP-tagged YTHDC2[H327R] showed a modest reduction of specific activity in the RNA unwinding assay (*Figure 9F*, *Figure 9—figure supplement 1B*). Thus, it is likely that functional inactivation caused by the H327R substitution is a consequence of misfolding or protein aggregation, which would be consistent with the observed decrease in protein levels in vivo (*Figure 7B*).

## Evolutionary path of ancestral *Ythdc2* and its divergent paralog, Drosophila *bgcn*

YTHDC2 homologs were found in metazoan species in a limited analysis of conservation (*Soh et al., 2017*). The closest homolog in *Drosophila melanogaster* is Bgcn (benign gonial cell neoplasm), which regulates germ cell differentiation via translational control of target mRNAs (*Li et al., 2009b*; *Kim et al., 2010*; *Insco et al., 2012*; *Chen et al., 2014*). Bgcn was thus proposed to be the fruit fly ortholog of YTHDC2 (*Soh et al., 2017*). However, Bgcn lacks a YTH domain and a clear match to the

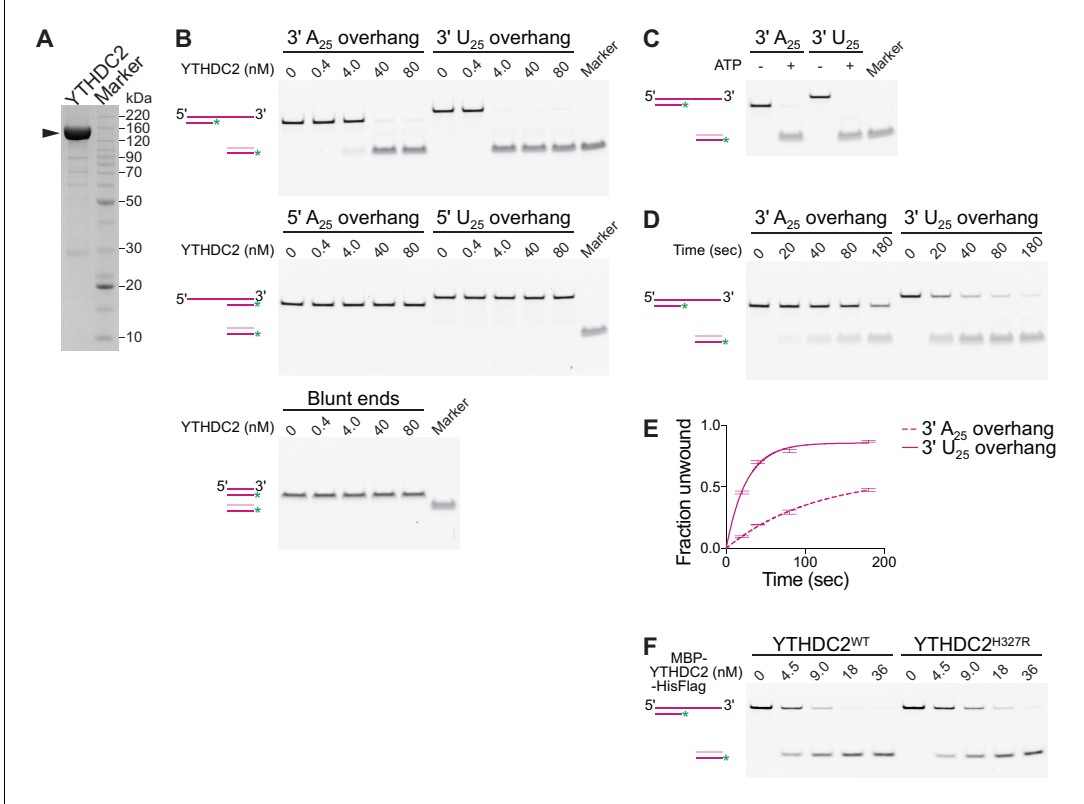

**Figure 9.** Mouse YTHDC2 is an ATP-dependent 3′→5′ RNA helicase. (**A**) SDS-PAGE analysis of recombinant mouse YTHDC2 (arrowhead) expressed in insect cells. (**B**) Strand displacement assay on 3′-tailed, 5′-tailed or blunt RNA duplexes. (**C**) Dependence of YTHDC2 helicase activity on ATP. In panels B and C, end point unwinding reactions were carried out at 22°C for 30 min. Marker lanes depict the expected band for the unwound 6-FAM-labeled RNA strand annealed to the DNA trap. (**D** and **E**) Representative native PAGE (panel D) and quantification (panel E) of time courses of unwinding reactions on 3′ $A_{25}$ and 3′ $U_{25}$ tailed RNA duplexes at 22°C using identical protein concentration (40 nM). In panel E, error bars indicate standard deviation from three technical replicates. Curves were fitted to the integrated pseudo-first order rate law to calculate observed rate constants ($k_{obs}$): 0.009 $sec^{-1}$ for the 3′ poly(A) substrate and 0.04 $sec^{-1}$ for the 3′ poly(U) substrate. (**F**) Comparison of helicase activities of wild-type YTHDC2 and YTHDC2[H327R] on 3′ $U_{25}$ tailed RNA duplex. The recombinant proteins used contain an N-terminal MBP tag to improve the solubility of YTHDC2[H327R] and a C-terminal hexahistidine tag to allow purification. End point unwinding reactions were carried out at 30°C for 30 min. For helicase assays, a schematic of the substrate (dark pink) before and after the reaction is depicted with the DNA trap shown in light pink and the 6-FAM label shown as a green asterisk.

DOI: https://doi.org/10.7554/eLife.30919.029

The following figure supplement is available for figure 9:

**Figure supplement 1.** Protein expression and helicase activity of recombinant YTHDC2[H327R].

DOI: https://doi.org/10.7554/eLife.30919.030

R3H domain (*Figure 10A*) and is altered in motifs critical for ATP binding and hydrolysis (*Ohlstein et al., 2000*) (*Figure 10B*). These differences led us to hypothesize either that the YTHDC2 family is highly plastic evolutionarily, or that YTHDC2 and Bgcn are not orthologs. To distinguish between these possibilities, we characterized the YTHDC2 phylogenetic distribution.

Likely YTHDC2 orthologs were readily found throughout the Eumetazoa subkingdom by BLAST and domain architecture searches (*Figure 10C*, *Figure 10—figure supplement 1* and *Supplementary file 3*). The complete YTHDC2 domain architecture (R3H, helicase core domains with intact helicase motifs I and II and ARD insertion, HA2, OB, and YTH domains) was the most common form (*Figures 8B* and *10A,C* and *Figure 10—figure supplement 2*). Orthologs of this type were present in Cnidaria, the deepest branching eumetazoans examined (*Hydra vulgaris* and the starlet sea anemone *Nematostella vectensis*) (*Figures 8B* and *10C*, *Figure 10—figure supplement*

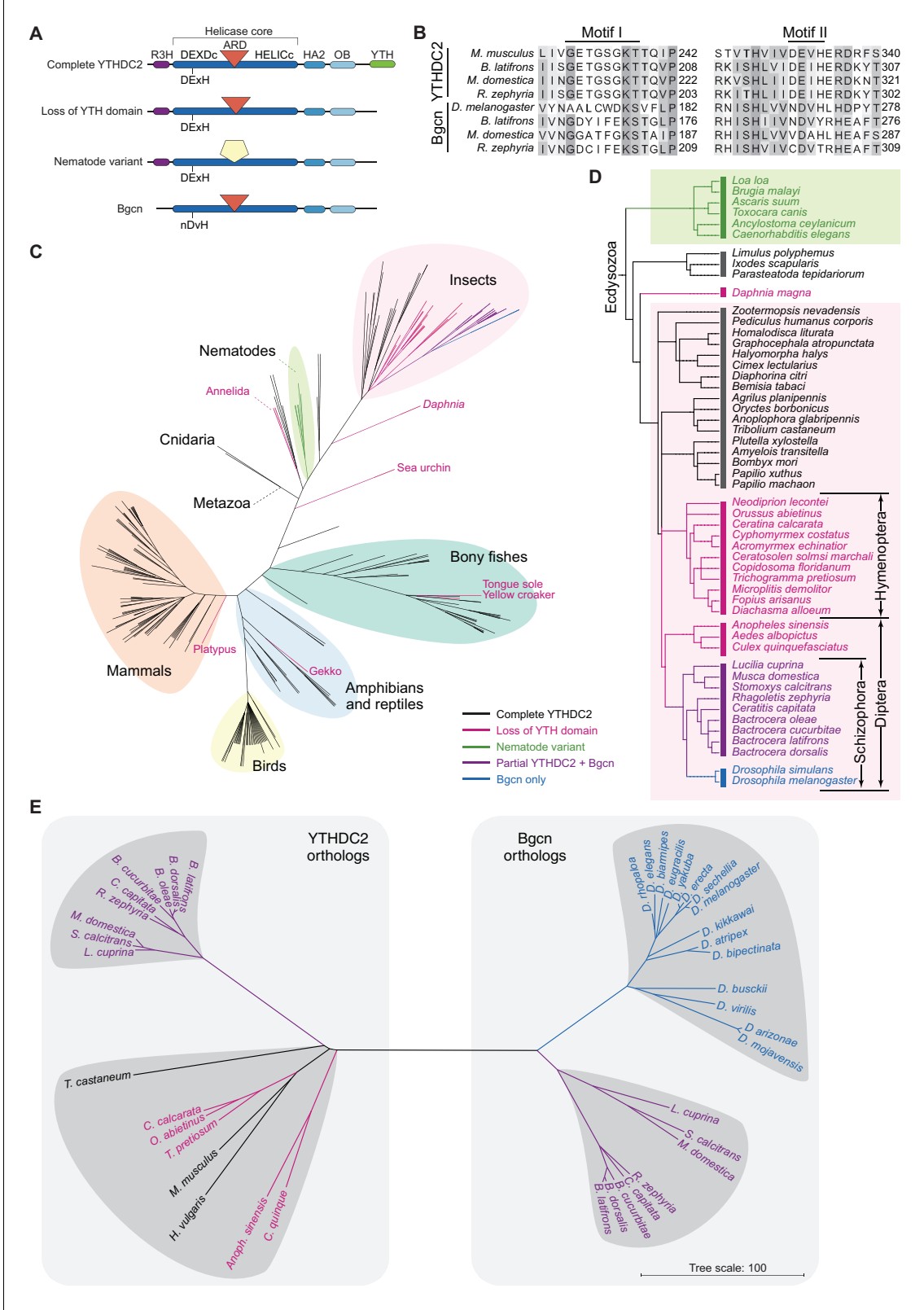

**Figure 10.** Distribution of YTHDC2 variants in Metazoa. (**A**) Schematic of domain architectures (not to scale) of metazoan YTHDC2 orthologs and paralogs. (**B**) Clustal Omega alignment of sequences around helicase motifs I and II for YTHDC2-like and Bgcn-like proteins from schizophoran flies. Mouse YTHDC2 is shown for comparison. Bgcn proteins have amino acid changes that are incompatible with ATPase activity (***Ohlstein et al., 2000***). (**C**) Phylogenetic distribution of YTHDC2 in Metazoa. The tree is an unrooted cladogram of NCBI taxonomy for a non-exhaustive collection of 234 species

*Figure 10 continued on next page*

*Figure 10 continued*

in which at least one close YTHDC2 homolog was identified. Tree leaves are color coded according to YTHDC2 domain architecture. The same tree topology is reproduced in *Figure 10—figure supplement 1* with complete species names. (D) Distribution of YTHDC2 variants in Ecdysozoa. The rooted cladogram shows NCBI taxonomy for the ecdysozoan portion of the metazoan tree in panel C. Background shading and color-coding of tree leaves is the same as in panel C. (E) Phylogram for sequence alignments of complete YTHDC2 and Bgcn orthologs from the indicated species. Note that protein sequence distances are similar within the YTHDC2 and Bgcn subfamily trees, but species in the YTHDC2 tree span much greater evolutionary distances. Sequences were aligned with Clustal Omega and the unrooted neighbor-joining tree was constructed from distances calculated using the scoredist function (*Sonnhammer and Hollich, 2005*). Tree leaves are color coded by YTHDC2 protein domain architecture as in panel C. Protein accession numbers are in *Supplementary file 3*.

DOI: https://doi.org/10.7554/eLife.30919.031

The following figure supplements are available for figure 10:

**Figure supplement 1.** Phylogenetic distribution of YTHDC2 in Metazoa.

DOI: https://doi.org/10.7554/eLife.30919.032

**Figure supplement 2.** Full length alignment of YTHDC2 orthologs and paralogs.

DOI: https://doi.org/10.7554/eLife.30919.033

**Figure supplement 3.** The YTHDC2 ortholog in nematodes has a distinct and highly diverse sequence in the location equivalent to the ARD insertion in other species.

DOI: https://doi.org/10.7554/eLife.30919.034

*1* and *Figure 10—figure supplement 2*), indicating that full-length YTHDC2 was present in the LCA of the Eumetazoa.

Homologs with the full domain architecture, except for the YTH domain, were observed in green plants, exemplified by *Arabidopsis thaliana* NIH (nuclear DEIH-box helicase) (*Isono et al., 1999*), and choanoflagellates, e.g., *Salpingoeca rosetta* (Genbank accession XP_004993066). No homologs with this domain architecture were found in other eukaryotic lineages, including fungi (see Materials and methods). The function of the plant homologs has not been reported to our knowledge, but their existence suggests an even more ancient evolutionary origin for this protein family.

Most metazoan YTHDC2 orthologs are highly conserved; for example, the mouse and *H. vulgaris* proteins are 44.6% identical. Nonetheless, there were exceptions to this conservation, including apparent sporadic loss of the YTH domain in some species (e.g., platypus and Japanese gekko among vertebrates; and Annelida (e.g., the leech *Helobdella robusta*) and Echinodermata (the purple sea urchin *Strongylocentratus purpuratus*) among invertebrates) (*Figure 10C* and *Figure 10—figure supplement 1*). Assuming these losses are not errors in genome assembly or annotation, these findings suggest that the mode of YTHDC2 binding to RNA can be evolutionarily plastic.

Even more diversity was observed in the superphylum Ecdysozoa, which includes nematodes and arthropods (*Figure 10C,D* and *Figure 10—figure supplement 1*). Most of these species retain the full architecture with the YTH domain, e.g., the horseshoe crab *Limulus polyphemus*; the common house spider *Parasteatoda tepidariorum*; the deer tick *Ixodes scapularis*; and most insect lineages. However, the YTH domain was apparently lost in at least one crustacean (*Daphnia magna*) and more widely in nematodes and in dipteran and hymenopteran insects. We examined these latter exceptions in more detail.

## Nematodes

All of the identified YTHDC2 homologs in the phylum Nematoda have a form distinct from other lineages: they retain the R3H, helicase core (including intact helicase motifs I and II), HA2, and OB domains, but they lack a detectable YTH domain. More uniquely, they have a sequence between the DEXDc and HELICc helicase core domains that aligns poorly with the equivalent region (including the ARD) in other family members, and that is more divergent between nematode species than between most other species (*Figures 8B* and *10A,D*, *Figure 10—figure supplement 2* and *Figure 10—figure supplement 3*). For example, the mouse and zebra finch (*Taeniopygia guttata)* proteins are 82.3% identical across this region, whereas the *Caenorhabditis elegans* and *Caenorhabditis briggsae* proteins share only 49.7% identity (*Figure 10—figure supplement 3B*). If the ARD of YTHDC2 mediates protein-protein interactions, the diverged structure of this region suggests that the protein-binding partners have also diverged substantially in nematodes.

## Insects

Most insect lineages have YTHDC2 orthologs with the full architecture including the YTH domain and intact helicase motifs I and II (*Figures 8B* and *10C,D*, *Figure 10—figure supplement 1* and *Figure 10—figure supplement 2*). Examples include Coleoptera (e.g., the red flour beetle *Triboleum castaneum*), Lepidoptera (e.g., the silk moth *Bombyx mori*), and Hemiptera (true bugs, e.g., the bed bug *Cimex lectularius*). We conclude that the ancestral metazoan form was present in the LCA of insects.

However, likely orthologs in Hymenoptera (e.g., bees, wasps, ants) and Diptera (e.g., flies, mosquitoes) uniformly lack the YTH domain, suggesting this domain was lost in the LCA of these two clades. In all Hymenoptera and some Diptera examined (mosquitoes), this YTH-less form was the only close homolog of YTHDC2 (*Figure 10D*).

Remarkably, an even more diverged homolog was found specifically in members of the Schizophora section of true flies (*Figure 10D*). Every species that we examined in this clade has a YTHDC2 homolog annotated as an ortholog of *D. melanogaster* Bgcn (*Figure 10E* and *Supplementary file 3*). Each has the ARD diagnostic of the YTHDC2 family but, like Bgcn, lacks the YTH domain and has Bgcn-like sequence alterations in helicase motifs I and II that are expected to preclude ATP binding and hydrolysis (*Ohlstein et al., 2000*) (*Figure 10A,B*, and *Figure 10—figure supplement 2*). Also like Bgcn, the N-terminal sequence of these proteins aligns poorly if at all with the R3H domain of other YTHDC2 proteins, indicating that this domain has been lost or is substantially divergent as well (*Figure 10—figure supplement 2*). In addition to this protein, most schizophoran flies also have a second YTHDC2 homolog that lacks the YTH domain but that, unlike Bgcn, has intact R3H and helicase motifs I and II (e.g., the house fly *Musca domestica* and the tephritid fruit flies *Rhagoletis zephyria*, *Ceratitus capetata*, and *Bactrocera* species; *Figure 10B,D,E* and *Figure 10—figure supplement 2*). None of the Drosophila genomes examined had this more YTHDC2-like version (*Figure 10D,E*).

A straightforward interpretation is that the YTH domain was lost before the LCA of Hymenoptera and Diptera, then a gene duplication before the LCA of Schizophora was followed by substantial sequence diversification, creating the Bgcn subfamily. Most schizophoran species have both YTHDC2-like and Bgcn-like paralogs, but Drosophilidae retain only the Bgcn version. Thus, although Bgcn is the closest YTHDC2 homolog in *D. melanogaster*, Bgcn is a paralog of YTHDC2, not its ortholog.

Supporting this interpretation, a phylogram based on multiple sequence alignments placed one version from each of the non-Drosophila schizophoran species closer to YTHDC2 orthologs from other species, including mouse, and placed the other copy closer to *D. melanogaster* Bgcn (*Figure 10E*). However, Bgcn paralogs from Drosophila and non-Drosophila species formed two distinct clusters, and the more YTHDC2-like schizophoran paralogs formed a cluster separated from other YTHDC2 members, including other insect proteins lacking a YTH domain (fuchsia branches in *Figure 10E*). Relative to YTHDC2 from non-schizophoran species, there is greater diversity between Bgcn family members within Schizophora and even within Drosophila (*Figure 10E*). Rapid evolution of Bgcn in Drosophila was noted previously, with evidence of positive selection in *D. melanogaster* and *D. simulans* (*Civetta et al., 2006*; *Bauer DuMont et al., 2007*) but not *D. ananassae* (*Choi and Aquadro, 2014*). Our findings extend this diversity to other schizophoran flies and place the evolutionary dynamics of Bgcn (and schizophoran YTHDC2) in striking contrast to the conservation of the ancestral YTHDC2 form in most other metazoan lineages.

## Drosophila *bag of marbles* encodes a highly diverged homolog of MEIOC

Like YTHDC2, MEIOC is highly conserved, with likely orthologs throughout most lineages in Eumetazoa, including Cnidaria (*Abby et al., 2016*; *Soh et al., 2017*) (*Figure 11A*). The MEIOC C-terminus contains a domain with a putative coiled-coil motif (DUF4582 (domain of unknown function); pfam15189); this domain is necessary and sufficient for interaction with YTHDC2 (*Abby et al., 2016*). In Eumetazoa, DUF4582 appears to be unique to MEIOC orthologs and is the protein's most highly conserved feature.

In *D. melanogaster*, the product of the *bag of marbles* (*bam*) gene is a functional collaborator and direct binding partner of Bgcn (*Gönczy et al., 1997*; *Lavoie et al., 1999*; *Ohlstein et al., 2000*; *Li et al., 2009b*; *Shen et al., 2009*; *Kim et al., 2010*; *Insco et al., 2012*; *Chen et al., 2014*). By

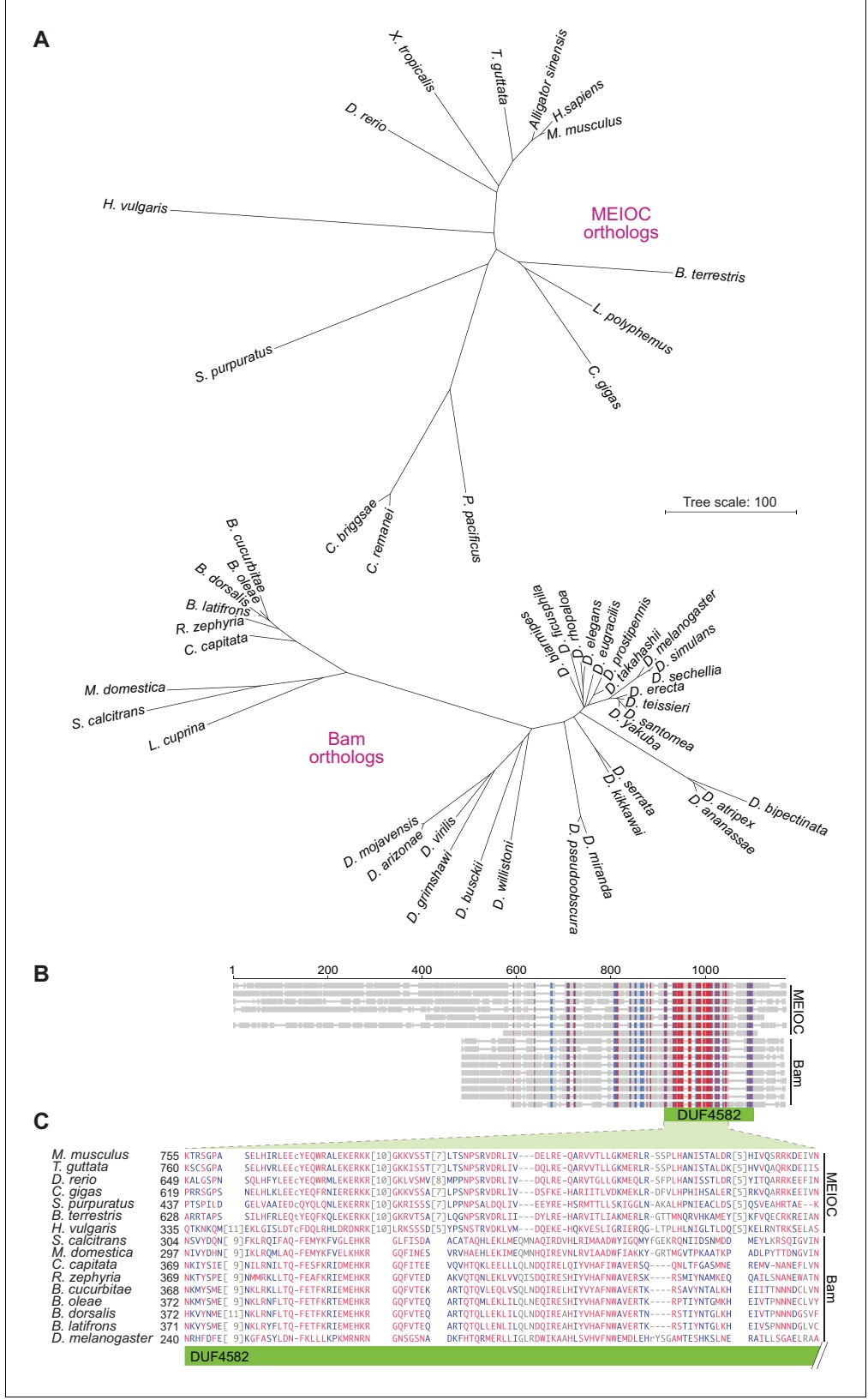

**Figure 11.** Bam shares distant sequence similarity with MEIOC. (**A**) Phylograms based on sequence alignments of MEIOC or Bam orthologs. Sequences were aligned with Clustal Omega and the unrooted neighbor-joining trees were constructed from distances calculated using the scoredist function. Note that the two trees have the same

*Figure 11 continued on next page*

*Figure 11 continued*

scale, but the MEIOC proteins are from species separated by much greater evolutionary distances. (**B** and **C**) Remote sequence similarity between MEIOC and Bam, concentrated across the conserved DUF4582 domain of MEIOC. Sequences were aligned using COBALT (*Papadopoulos and Agarwala, 2007*). Panel B shows a schematic of the full alignment, with thin gray lines indicating gaps and thick gray lines indicating amino acid sequence. Species are in the same order as panel C, which shows a zoomed-in view of the region of greatest contiguous sequence similarity. Residues aligned across all proteins with no gaps are colored in blue or red according to relative entropy, with red indicating more highly conserved (entropy threshold = 2 bits) (https://www.ncbi.nlm.nih.gov/tools/cobalt/re_cobalt.cgi). Boundaries of DUF4582 are indicated relative to their annotation in the mouse protein. Protein accession numbers are in *Supplementary file 3*.
DOI: https://doi.org/10.7554/eLife.30919.035

analogy with YTHDC2-MEIOC, it was thus proposed that Bam may be a functional analog of MEIOC (*Soh et al., 2017*). However, no sequence similarity between MEIOC and Bam has been detected. Moreover, the rapid diversification of Bgcn and in particular its biochemical divergence from the ancestral YTHDC2 form raised the possibility that Bgcn has acquired novel interaction partners, i.e., that Bam and MEIOC are not evolutionarily related. To address these issues, we examined the sequence and phylogenies of Bam and MEIOC.

BLAST searches using mouse or human MEIOC as queries against available dipteran genomes identified no clear homologs (expected (E) value threshold <20), even though homologs containing DUF4582 were easily found in other insect lineages (E < $10^{-35}$), including Hymenoptera (e.g., the bumblebee *Bombus terrestris*) (*Figure 11A*, *Supplementary file 3*). Conversely, searches using *D. melanogaster* Bam as the query identified homologs in schizophoran flies (E < $10^{-3}$; *Figure 11A*, *Supplementary file 3*), establishing that Bam orthologs are coincident with the presence of Bgcn-like proteins. However, these searches failed to find Bam homologs in any other species, including non-schizophoran Diptera.

Nonetheless, evidence of remote sequence similarity was observed when MEIOC orthologs from widely divergent species were compared directly with schizophoran Bam orthologs by multiple sequence alignment using COBALT (constrained basic alignment tool [*Papadopoulos and Agarwala, 2007*]) (*Figure 11B,C*) or PROMALS3D ([*Pei and Grishin, 2014*]; data not shown). Bam orthologs are shorter, lacking an N-terminal extension present in MEIOC (*Figure 11B*). Short patches of sequence similarity with Bam were distributed across the central, nondescript region of MEIOC, but the region with greatest similarity spanned much of the DUF4582 domain (*Figure 11B*). Supporting the significance of this similarity, the C-terminus of Bam, including part of the conserved region, mediates the direct interaction with Bgcn (*Li et al., 2009b*), as DUF4582 does for MEIOC-YTHDC2 (*Abby et al., 2016*). Interestingly, however, the COILS prediction algorithm (*Lupas et al., 1991*) did not detect putative coiled-coil motifs in Bam, unlike MEIOC (data not shown). We conclude that *D. melanogaster* Bam is evolutionarily derived from MEIOC and has a functionally homologous version of the DUF4582 domain, albeit diverged enough that it is not readily recognized as such.

A neighbor-joining tree based on multiple sequence alignments divided schizophoran Bam-like proteins into two clusters representing, respectively, Drosophila and non-Drosophila species (*Figure 11A*). Furthermore, Bam-like proteins showed substantially more sequence diversity than MEIOC-like proteins, and there also was more diversity within Drosophila species than within the other schizophoran flies (*Figure 11A*). Thus, conservation patterns are correlated between MEIOC/Bam and YTHDC2/Bgcn: YTHDC2 and MEIOC are much more highly conserved than are Bgcn and Bam, and Bgcn and Bam display even greater sequence diversity among Drosophila species than in other clades in Schizophora.

## Discussion

This study establishes an essential function for *Ythdc2* in the germlines of male and female mice, specifically at the stage when stem cells transition from mitotic to meiotic divisions. *Ythdc2* mutant spermatogonia are able to initiate at least part of the meiotic developmental program, i.e., expressing meiosis-specific transcripts, making synaptonemal complex precursors, and initiating recombination. However, most cells then rapidly progress to a premature metaphase-like state and die by

apoptosis. *Ythdc2* and *Meioc* mutants have highly similar meiotic phenotypes, supporting the hypothesis that these proteins function together to regulate germ cell differentiation in the mouse gonad (*Abby et al., 2016*; *Soh et al., 2017*). During the course of this work, we isolated a novel point-mutated allele of *Ythdc2* (*ketu*) that harbors a non-synonymous mutation of a conserved residue, illustrating the power of phenotype-based forward-genetic approaches for dissecting mammalian reproductive processes.

Our findings agree well with recent independent studies of *Ythdc2* function published while our manuscript was under review (*Bailey et al., 2017*; *Hsu et al., 2017*; *Wojtas et al., 2017*). In broad strokes, each study of *Ythdc2* and of *Meioc* has led to the same general conclusion that the YTHDC2-MEIOC complex regulates the switch from mitotic divisions to meiosis (this work and [*Abby et al., 2016*; *Soh et al., 2017*; *Bailey et al., 2017*; *Hsu et al., 2017*; *Wojtas et al., 2017*]). STRA8 is a retinoic acid-induced factor that also regulates the mitosis to meiosis switch (*Baltus et al., 2006*; *Anderson et al., 2008*). Interestingly, spermatocytes lacking *Stra8* have been reported to undergo an abortive entry into meiosis with premature progression to a metaphase-like state similar to *Ythdc2* or *Meioc* mutants (*Mark et al., 2008*), although other studies have observed a more complete block to meiotic entry in $Stra8^{-/-}$ mice (*Baltus et al., 2006*; *Anderson et al., 2008*), possibly reflecting variation attributable to strain background. On the basis of possible shared features between $Stra8^{-/-}$ and $Meioc^{-/-}$ mutants, it was proposed that MEIOC (and by inference YTHDC2) may overlap functionally with STRA8 in realizing the complete meiotic program (*Abby et al., 2016*). Our findings are consistent with this hypothesis.

The YTHDC2 domain architecture, with its RNA interaction modules and putative RNA helicase domains, as well as the cytoplasmic localization of the YTHDC2-MEIOC complex (this study and [*Abby et al., 2016*; *Bailey et al., 2017*; *Soh et al., 2017*]) leads to the obvious hypothesis that YTHDC2 regulates gene expression post-transcriptionally via direct interaction with specific RNA targets. Precisely how this regulation is accomplished remains unclear, however. Two distinct models have been proposed in which the principal function of YTHDC2-MEIOC is to control mRNA stability, either stabilizing the transcripts of meiotic genes (*Abby et al., 2016*) or destabilizing the transcripts of mitotic cell cycle genes (*Hsu et al., 2017*; *Soh et al., 2017*; *Wojtas et al., 2017*). Our RNA-seq data suggest that both possibilities are correct, i.e., that YTHDC2-MEIOC has (directly or indirectly) opposite effects on the steady-state levels of different classes of transcripts. If so, the molecular determinants of these context-dependent outcomes remain to be elucidated. The $5' \rightarrow 3'$ exoribonuclease XRN1 was proposed to be a functional partner of the YTHDC2-MEIOC complex on the basis of anti-MEIOC immunoprecipitation and mass spectrometry (IP/MS) from testis extracts (*Abby et al., 2016*). More recent work supported this hypothesis and suggested that the ARD of YTHDC2 mediates the interaction with XRN1 (*Hsu et al., 2017*; *Wojtas et al., 2017*), although evidence of direct contact between XRN1 and the ARD is not yet available.

It has also been proposed that YTHDC2 promotes mRNA translation, based on effects of shRNA knockdown of *YTHDC2* in human colon cancer cell lines (*Tanabe et al., 2016*), as well as measures of apparent translation efficiency of putative YTHDC2 targets in mouse testis and of artificial reporter constructs in HeLa cells (*Hsu et al., 2017*). An alternative possibility is that YTHDC2-MEIOC has translational suppressor activity based on analogy with Bgcn-Bam (*Bailey et al., 2017*), which has well documented translational inhibition functions in *D. melanogaster* germ cells (*Li et al., 2009b*; *Shen et al., 2009*; *Kim et al., 2010*; *Insco et al., 2012*; *Li et al., 2013*; *Chen et al., 2014*). Interestingly, MEIOC and/or YTHDC2 IP/MS experiments have also identified a number of proteins with roles in translation (*Abby et al., 2016*; *Hsu et al., 2017*; *Wojtas et al., 2017*). These interactions remain to be validated, but are consistent with possible roles of the YTHDC2-MEIOC complex in translational control. Finally, we note that control of transcript stability and translation are not mutually exclusive, and the cytoplasmic localization of the YTHDC2-MEIOC complex (this study and [*Abby et al., 2016*; *Bailey et al., 2017*; *Soh et al., 2017*]) is consistent with all of these possibilities.

RNA-stimulated ATPase activity was first reported for purified YTHDC2 by *Morohashi et al. (2011)* and subsequently confirmed by *Wojtas et al. (2017)*. We and *Wojtas et al., 2017* observed ATP-dependent RNA unwinding activity on substrates with a 3'-single-stranded extension, consistent with a $3' \rightarrow 5'$ polarity. The role this helicase activity plays in YTHDC2 function in vivo remains to be deciphered. Possible precedents come from other mammalian DEAH helicases that are involved in mRNA translation and stability, including DHX29 (which promotes translation initiation by unwinding highly structured 5' untranslated regions to allow efficient 48S complex formation on mRNAs

[*Pisareva et al., 2008*]) and DHX36 (which cooperates with the PARN deadenylase and the RNA exosome for degradation of mRNAs containing an AU-rich element [*Tran et al., 2004*]).

RNA co-immunoprecipitation data suggest that YTHDC2 interacts with specific RNA targets in vivo (*Abby et al., 2016*; *Bailey et al., 2017*; *Hsu et al., 2017*; *Soh et al., 2017*), but the detailed lists of putative targets have differed between studies and the molecular determinants of binding specificity remain unknown. Structural analyses support the conclusion that the YTH domain mediates direct interaction with m$^6$A-containing RNA substrates (this study and [*Wojtas et al., 2017*]), also suggested by direct binding studies in vitro (*Xu et al., 2015*; *Hsu et al., 2017*; *Wojtas et al., 2017*). Our in vitro studies also indicate functional discrimination between RNAs of different sequence, e.g., a possible preference for RNA with poly-uridine sequences. Binding to multiple sequence elements may provide more specificity and limit promiscuity in substrate recognition. The various domains in YTHDC2 may confer multivalent interaction with nucleic acid and dictate different molecular outcomes depending on which domains engage the RNA.

The biochemical role of MEIOC within the complex is also unknown. Non-exclusive possibilities include modulating YTHDC2 RNA-binding specificity, facilitating interactions with other protein partners, and/or modifying ATPase/helicase activities of YTHDC2. Moreover, notwithstanding the intimate connection of YTHDC2 to MEIOC in the germline, it is likely that YTHDC2 has additional, independent functions, because it is expressed much more widely than the highly germline-specific MEIOC. Supporting this hypothesis, YTHDC2 has been implicated in hepatitis C virus replication and cell proliferation in cultured, transformed human cells not known to express MEIOC (*Morohashi et al., 2011*; *Tanabe et al., 2014*; *Tanabe et al., 2016*). Although we have not observed obvious somatic phenotypes in *Ythdc2*$^{ketu/ketu}$ or *Ythdc2*$^{em1/em1}$ homozygotes, we cannot rule out cellular defects that do not yield gross pathology.

We demonstrate here that the YTHDC2 sequence is well conserved across most metazoan lineages, including the deeply branching Cnidaria. Hence, we conclude that the full-length YTHDC2 is the ancestral form, already present in the LCA of Metazoa. A related protein (possibly lacking the YTH domain) was likely present even earlier, before the LCA of green plants and metazoans. However, we also uncovered substantial structural diversity in the nematode, hymenopteran, and dipteran lineages within Ecdysozoa, and particularly in schizophoran flies. In the simplest cases (Hymenoptera and non-schizophoran Diptera), the structural variation consists principally of loss of the YTH domain. The YTH domain thus appears to be an evolutionarily and biochemically modular contributor to YTHDC2 function.

The nematode variant also lacks a YTH domain, but in addition, the region equivalent to the ARD is highly divergent relative to the ancestral sequence and even between different nematode species. It remains to be determined whether this diversification reflects positive selection for changing sequence (source unknown) or neutral selection (e.g., if the nematode protein no longer interacts with XRN1 or some other partner, relaxing constraint on the ARD sequence). The function of this protein in nematodes also remains unknown. The *C. elegans* ortholog of YTHDC2 is F52B5.3 (*Supplementary file 3*), and the MEIOC ortholog is Y39A1A.9 (*Abby et al., 2016*). Both proteins are poorly characterized and, to our knowledge, no phenotypes caused by mutation or RNAi have been observed (http://www.wormbase.org/).

More striking still, the dipteran YTH-less YTHDC2 family member in the LCA of the Schizophora appears to have been duplicated to form the Bgcn family, which also apparently lost the R3H domain and accumulated the previously described alterations in ATPase motifs that preclude ATP binding and hydrolysis (*Ohlstein et al., 2000*), thus making it impossible for Bgcn to have the ATP-dependent RNA helicase activity that is presumably retained in other YTHDC2 family members. Our studies also revealed for the first time that the Bgcn partner Bam is a divergent homolog of MEIOC unique to Schizophora. Because we have been unable to identify Bam/MEIOC homologs in non-schizophoran Diptera, it is currently unclear if *bam* arose from a *Meioc* gene duplication (analogous to the evolutionary trajectory of *bgcn*) or if it is simply a highly diverged *Meioc* ortholog. Given the apparent presence of only Bam-like proteins in non-Drosophila Schizophora, it would be interesting to know whether Bam interacts with both the Bgcn and YTHDC2 orthologs in these species.

Previous researchers documented that both Bam and Bgcn are rapidly diversifying in Drosophila species (*Civetta et al., 2006*; *Bauer DuMont et al., 2007*; *Choi and Aquadro, 2014*). Our findings extend this property to schizophoran flies more generally and provide further context by showing that both Bgcn-like and, when present, YTHDC2-like proteins have experienced much greater

sequence diversification within Schizophora than elsewhere, and that this is mirrored by more rapid sequence changes in Bam compared with the ancestral MEIOC in other lineages. The coincident occurrence — at or before the LCA of Schizophora — of the *Ythdc2/bgcn* gene duplication, the YTHDC2, Bgcn, and Bam diversification, and the Bgcn structural and biochemical changes makes it tempting to speculate that these were coordinate changes driven by a common set of selective pressures.

If so, what was (is) the source of these pressures? It was speculated that the diversification of Bam and Bgcn may be tied to infection with the alpha-proteobacteria *Wolbachia* (*Bauer DuMont et al., 2007*). *Wolbachia* is an endosymbiont in many species of insects and other arthropods that is transmitted from parent to offspring and that manipulates host reproductive processes (*Engelstädter and Hurst, 2009*; *Pietri et al., 2016*). Interestingly, *Wolbachia* also infects a number of nematode species (*Ferri et al., 2011*), suggesting a possible link to the rapid diversification of YTHDC2 orthologs in that clade as well. However, direct evidence in any species for a link between Bgcn/Bam and *Wolbachia* remains elusive. Moreover, many arthropod species across diverse taxa are infected with *Wolbachia* (*Engelstädter and Hurst, 2009*; *Pietri et al., 2016*; *Rosenfeld et al., 2016*), but we find that the majority of these taxa have more evolutionarily stable YTHDC2 and MEIOC sequences. Thus, if Bgcn and Bam diversification can be attributed to host genomic conflicts with *Wolbachia*, it may reflect a mode of interaction between *Wolbachia* and the germline that is unique to schizophoran flies (and possibly nematodes).

The complex evolutionary relationships we document here raise the possibility that mammalian YTHDC2 and fly Bgcn have substantially different functions and mechanisms of action, especially given the striking biochemical changes private to the Bgcn subfamily. Moreover, the phenotypic outcomes in mutants differ, for example *bam* and *bgcn* mutant germ cells do not enter meiosis (*McKearin and Spradling, 1990*; *Gönczy et al., 1997*), whereas *Meioc* and *Ythdc2* mutants do. Nonetheless, our findings and others' (*Bailey et al., 2017*; *Hsu et al., 2017*; *Wojtas et al., 2017*), along with the recent characterization of *Meioc* mutants (*Abby et al., 2016*; *Soh et al., 2017*), also establish intriguing parallels with *bgcn* and *bam*: in both mouse and fruit fly these genes play critical roles in the switch from mitotic cell cycles into meiosis, in both male and female germlines. On the basis of this similarity, and because members of both the YTHDC2/Bgcn and MEIOC/Bam families appear to be nearly ubiquitous in metazoans, we propose that the YTHDC2-MEIOC complex has an evolutionarily ancient and conserved function as a regulator of germ cell fate and differentiation.

## Materials and Methods

### Generation of *ketu* mutants and endonuclease-targeted *Ythdc2* mutations

All experiments conformed to regulatory standards and were approved by the Memorial Sloan Kettering Cancer Center (MSKCC) Institutional Animal Care and Use Committee. Wild-type B6 and FVB mice were purchased from The Jackson Laboratory (Bar Harbor, Maine). Details of the ENU mutagenesis and breeding for screening purposes are provided elsewhere (*Jain et al., 2017*) and were similar to previously described methods (*Caspary, 2010*; *Probst and Justice, 2010*).

To screen for meiotic defects, spermatocyte squash preparation and immunostaining for SYCP3 and γH2AX (described below) were carried out using testes from pubertal G3 males that were ≥15 *dpp* or from adult G3 males. At these ages, spermatocytes in the relevant stages of meiotic prophase I are abundant in normal mice (*Bellvé et al., 1977*). Testes were dissected, frozen in liquid nitrogen, and stored at −80°C until samples from ~24 G3 males had been collected, then immunocytology was carried out side by side for all animals from a given line. One testis per mouse was used for cytology and the second was reserved for DNA extraction.

Genotyping of *ketu* animals was done by PCR amplification using *ketu* F and *ketu* R primers (oligonucleotide primer sequences are provided in *Supplementary file 4*), followed by digestion of the amplified product with *Bst*XI (NEB, Ipswich, Massachusetts). The wild-type sequence contains a *Bst*XI restriction site that is eliminated by the *ketu* mutation.

CRISPR/Cas9-mediated genome editing was done by the MSKCC Mouse Genetics Core Facility to generate *em* alleles. A guide RNA (target sequence 5′-AATAAAGGCTCTTTCCGTAC) was designed to target predicted exon 2 of *Ythdc2* (NCBI Gene ID: 240255 and Ensembl Gene ID:

ENSMUSG00000034653) and used for editing as described (*Romanienko et al., 2016*). Using the T7 promoter in the pU6T7 plasmid, the gRNA was synthesized by in vitro transcription and polyadenylated, then 100 ng/μl of gRNA and 50 ng/μl of Cas9 mRNA were co-injected into the pronuclei of CBA × B6 F2 hybrid zygotes using conventional techniques (*Hogan and Lacy, 1994*). Founder mice were tested for the presence of mutated alleles by PCR amplification of exon 2 using *Ythdc2* F1 and *Ythdc2* R1primers, followed by T7 endonuclease I (NEB) digestion.

To determine the specific mutations in T7-positive *Ythdc2*^em founder mice, the targeted region was amplified by PCR of tail-tip DNA (*Ythdc2* F1 and *Ythdc2* R1 primers) and sequenced on the Illumina MiSeq platform (Illumina Inc, San Diego, California) at the MSKCC Integrated Genomics Operation. Reads were aligned to mouse genome assembly GRCm37/mm9 and variants were identified using Genome Analysis Toolkit version 2.8-1-g932cd3a (*McKenna et al., 2010*; *DePristo et al., 2011*; *Van der Auwera et al., 2013*). Variants with a minimum variant frequency of 0.01 were annotated using VarScan v2.3.7 software (*Koboldt et al., 2012*). *Ythdc2*^em founder males mosaic for frame-shift mutations were bred with B6 mice and potential heterozygote carriers were genotyped by PCR amplification of the targeted region (*Ythdc2* F2 and *Ythdc2* R2 primers), followed by Sanger sequencing (*Ythdc2* Seq1 primer) and analysis with CRISP-ID (*Dehairs et al., 2016*). A single founder male carrying the *em1* mutation was used to establish the *Ythdc2*^em1 line, after two backcrosses to B6 mice. *Ythdc2*^em1 heterozygote carriers were then interbred to generate homozygotes, or crossed to *ketu* mice to generate compound heterozygotes carrying both the *ketu* allele and a *Ythdc2*^em1 allele. Genotyping of *Ythdc2*^em1 animals was done by PCR amplification using *Ythdc2* F3 and *Ythdc2* R3 primers, followed by digestion of the amplified product with *Rsa*I (NEB). The 5 bp deletion in *Ythdc2*^em1 removes an *Rsa*I site that is present in the wild-type sequence. Mouse Genome Informatics (MGI) accession numbers for *Ythdc2* alleles in this study are MGI:5911552 (*Ythdc2*^ketu) and MGI:5911553 (*Ythdc2*^em1Sky).

## Genetic mapping and exome sequencing

Genome assembly coordinates are from GRCm38/mm10 unless indicated otherwise. *ketu* was coarsely mapped by genome-wide microarray SNP genotyping (Illumina Mouse Medium Density Linkage Panel) using genomic DNA from testes or tail biopsies as described (*Jain et al., 2017*). Five G3 mutant mice obtained from the initial screen cross (a, b, c, d, e in *Figure 1C*), as well as the F1 founder, one B6, and one FVB control mice were genotyped. Microarray analysis was performed at the Genetic Analysis Facility, The Centre for Applied Genomics, The Hospital for Sick Children, Toronto, ON, Canada. For bioinformatics analysis, 777 SNPs were selected based on the following criteria: allelic variation in B6 and FVB, heterozygosity in F1 founder, and autosomal location.

We performed whole-exome sequencing on the same five mutant G3 mice analyzed by microarray SNP genotyping and DNA was prepared as for microarray analysis. Whole-exome sequencing was performed at the MSKCC Integrated Genomics Operation. A unique barcode was incorporated into the DNA library prepared from each mouse, followed by library amplification with 4 PCR cycles. Libraries were then quantified and pooled at equal concentrations into a single sample for exome capture. Exome capture was performed using SureSelectXT kit (Agilent Technologies, Santa Clara, California) and SureSelect Mouse All Exon baits (Agilent Technologies). Libraries were amplified post-capture with 6 PCR cycles and sequenced on the Illumina HiSeq platform to generate approximately 80 million 100 bp paired-end reads. Read alignment, variant calling and variant annotation were done as described (*Jain et al., 2017*), with the following two modifications. Reads with three or more mismatches and reads without a pair were discarded using SAMtools version 0.1.19 (*Li et al., 2009a*). Variants were filtered to only include those that had a minimum sequencing depth of five reads, were called as homozygous, and were not known archived SNPs.

## ENCODE data analysis

ENCODE long-RNA sequencing data (release 3) with the following GEO accession numbers were used: testis GSM900193, cortex GSM1000563, frontal lobe GSM1000562, cerebellum GSM1000567, ovary GSM900183, lung GSM900196, large intestine GSM900189, adrenal gland GSM900188, colon GSM900198, stomach GSM900185, duodenum GSM900187, small intestine GSM900186, heart GSM900199, kidney GSM900194, liver GSM900195, mammary gland GSM900184, spleen

GSM900197, thymus GSM900192, whole brain E14.5 GSM1000572, limb E14.5 GSM1000568, liver E14.5 GSM1000571.

## Histology

Testes from adult or juvenile mice were fixed overnight in 4% paraformaldehyde (PFA) at 4°C, or in Bouin's fixative for 4 to 5 hr at room temperature. Bouin's-fixed testes were washed in water for 1 hr at room temperature, followed by five 1 hr washes in 70% ethanol at 4°C. Wild-type and mutant ovaries were fixed in 4% PFA, overnight at 4°C or for 1 hr at room temperature, respectively. PFA-fixed tissues were washed twice for 5 min in water at room temperature. Fixed tissues were stored in 70% ethanol for up to 5 days prior to embedding, embedded in paraffin, and sectioned (4 or 5 µm). Periodic acid Schiff (PAS) staining, immunohistochemical TUNEL assay, and immunofluorescent staining were performed by the MSKCC Molecular Cytology Core Facility using the Autostainer XL (Leica Microsystems, Wetzlar, Germany) automated stainer for PAS with hematoxylin counterstain, and using the Discovery XT processor (Ventana Medical Systems, Oro Valley, Arizona) for TUNEL and immunofluorescent staining (*Yarilin et al., 2015*). TUNEL assay was performed as previously described (*Jain et al., 2017*). For immunofluorescent staining, slides were deparaffinized with EZPrep buffer (Ventana Medical Systems), antigen retrieval was performed with CC1 buffer (Ventana Medical Systems), and slides were blocked for 30 min with Background Buster solution (Innovex, Richmond, California). For staining of pH3, α-tubulin, CCNA2, YTHDC2, SYCP3 with CCNA2, and SYCP3 with YTHDC2, avidin-biotin blocking (Ventana Medical Systems) was performed for 8 min. Slides were incubated with primary antibody for 5 hr, followed by 60 min incubation with biotinylated goat anti-rabbit, horse anti-goat, or horse anti-mouse antibodies (1:200, Vector Labs, Burlingame, California). Streptavidin-HRP D (Ventana Medical Systems) was used for detection, followed by incubation with Tyramide Alexa Fluor 488 or 594 (Invitrogen, Carlsbad, California). SYCP3 with BrdU staining was done with the following modifications: avidin-biotin blocking was omitted and the detection step with Streptavidin-HRP D included Blocker D (Ventana Medical Systems). For BrdU staining, slides were blocked for 30 min with Background Buster solution and then treated with 5 µg/ml Proteinase K in Proteinase K buffer for 4 min. Slides were incubated with primary antibody for 3 hr, followed by 20 min treatment with 0.1% $H_2O_2$ in water, followed by 60 min incubation with biotinylated anti-mouse antibody. Detection was performed with Secondary Antibody Blocker (Ventana Medical Systems), Blocker D and Streptavidin-HRP D, followed by incubation with Tyramide Alexa Fluor 568 (Invitrogen). After staining, slides were counterstained with 5 µg/ml 4′,6-diamidino-2-phenylindole (DAPI) (Sigma, Darmstadt, Germany) for 10 min and mounted with coverslips with Mowiol.

PAS-stained and TUNEL slides were digitized using Pannoramic Flash 250 (3DHistech, Budapest, Hungary) with 20×/0.8 NA air objective. Images were produced and analyzed using the Pannoramic Viewer software (3DHistech). Higher magnification images of PAS-stained testis sections were produced using Axio Observer Z2 microscope (Carl Zeiss, Oberkochen, Germany) with 63×/1.4 NA oil-immersion objective. Most immunofluorescence images were produced using a TCS SP5 II confocal microscope (Leica Microsystems) with 40×/1.25 NA or 63×/1.4 NA oil-immersion objective. For immunofluorescence images in *Figure 4A* and *Figure 4—figure supplement 1A*, slides were digitized using Pannoramic Flash 250 with 40×/0.95 NA air objective and Pannoramic Confocal (3DHistech) with 40×/1.2 NA water-immersion objective, respectively, and images were produced using the CaseViewer 2.1 software (3DHistech).

## BrdU treatment

BrdU (Sigma) was administered into 15 *dpp*-old animals by two rounds of intraperitoneal injections, with a one-hour interval between the first and second injections. BrdU was dissolved in PBS and a dose of 50 µg/g of body weight was administered in a maximum volume of 70 µl each time. Testes were harvested two hours after the first injection and processed for immunofluorescent staining as described above.

## Histological examination of somatic tissues

Gross histopathological analysis of major organs and tissues was performed by the MSKCC Laboratory of Comparative Pathology Core Facility for the following male mice: two *Ythdc2^{ketu}* mutants plus one wild-type and one heterozygous littermates aged 10 weeks; one *Ythdc2^{ketu}* mutant and

one heterozygous littermate aged 65 weeks; one *Ythdc2^{ketu}* mutant and one wild-type littermate aged 67 weeks; one *Ythdc2^{em1}* mutant and one wild-type littermate aged 10 weeks; one *Ythdc2^{em1}* mutant and one heterozygous littermate aged 12 weeks; two *Ythdc2^{em1}* mutants plus one wild-type and one heterozygous littermates aged 43 weeks. The following females were analyzed: one *Ythdc2^{ketu}* mutant and one wild-type littermate aged 16 weeks; one *Ythdc2^{ketu}* mutant and one wild-type littermate aged 24 weeks; one *Ythdc2^{em1}* mutant and one wild-type littermate aged 32 weeks; one *Ythdc2^{em1}* mutant and one heterozygous littermate aged 34 weeks. Histologic examination of the following tissues was performed: diaphragm, skeletal muscle, sciatic nerve, heart/aorta, thymus, lung, kidneys, salivary gland, mesenteric lymph nodes, stomach, duodenum, pancreas, jejunum, ileum, cecum, colon, adrenals, liver, gallbladder, spleen, uterus, ovaries, cervix, urinary bladder, skin of dorsum and subjacent brown fat, skin of ventrum and adjacent mammary gland, thyroid, parathyroid, esophagus, trachea, stifle, sternum, coronal sections of head/brain, vertebrae and spinal cord. Tissues were fixed in 10% neutral buffered formalin and bones were decalcified in formic acid solution using the Surgipath Decalcifier I (Leica Biosystems, Wetzlar, Germany) for 48 hr. Samples were routinely processed in alcohol and xylene, embedded in paraffin, sectioned (5 µm), and stained with hematoxylin and eosin. Mutant males examined had marked degeneration of seminiferous tubules with aspermia, while in wild-type or heterozygous littermates, testes appeared normal. Mutant females examined had marked ovarian atrophy with afolliculogenesis, while in wild-type or heterozygous littermates, ovaries appeared normal. All other findings in mutants were considered incidental and/or age-related.

## Image analysis

Quantitative data presented in *Figure 1D*, *Figure 3—figure supplement 1*, *Figure 4D,E* were acquired by manual scoring. Acquisition of total cell counts in *Figure 4D* and quantitative data presented in *Figure 5* and *Figure 5—figure supplement 1* was automated. Slides were imaged using Pannoramic Flash 250 with 40×/0.95 NA air objective, then TIFF images of individual sections were exported using CaseViewer 2.1 and imported into Fiji software (*Schindelin et al., 2012*; *Rueden et al., 2017*) for analysis. The DAPI channel of images was thresholded (Bernsen's method) and segmented (Watershed method) to count and analyze individual cells. The average intensity of the FITC and TRITC channels were measured for each cell and exported as 8-bit intensity values. Average intensity values were corrected for background using MATLAB (MathWorks, Natick, Massachusetts) by subtracting the slope of the intensity values. Scatter plots were created using background-corrected 8-bit intensity values. The average intensity of ~10 cells judged to be positive for staining based on cell morphology and location within seminiferous tubules was measured and used to determine a minimum intensity threshold for scoring cells as positively stained.

## Cytology

Spermatocyte squashes were prepared as described (*Page et al., 1998*), with modifications as indicated in (*Jain et al., 2017*) and slides were stored at −80°C. Slides were thawed in 1× PBS for 5 min with gentle agitation and immunofluorescent staining of squashes was performed as described (*Dowdle et al., 2013*) using primary and appropriate Alexa Fluor secondary antibodies (1:100; Invitrogen). Primary antibody staining was done overnight at 4°C and secondary antibody staining was done for 30 min at room temperature. All antibodies were diluted in blocking buffer. Stained slides were rinsed in water and mounted with coverslips using mounting medium (Vectashield, Vector Labs) containing DAPI. Spermatocyte spreads were prepared as described (*Peters et al., 1997*) and immunofluorescent staining of spreads was performed as done for squashes with the following modifications: slides were treated with an alternative blocking buffer (1× PBS with 10% donkey serum, 0.05% TWEEN-20, 1% non-fat milk), primary antibody staining was done overnight at room temperature, and secondary antibody staining was done with appropriate Alexa Fluor secondary antibodies (1:500; Invitrogen) for 1 hr at 37°C. Slides with squashes or spreads were stored at 4°C for up to 5 days, and were imaged on a Marianas Workstation (Intelligent Imaging Innovations (Denver, Colorado); Zeiss Axio Observer inverted epifluorescent microscope with a complementary metal-oxide semiconductor camera) using a 63× oil-immersion objective.

## Antibodies

Primary antibodies and dilutions used for cytology are as follows: mouse anti-SYCP3 (SCP-3 (D-1), 2 µg/ml, Santa Cruz (Dallas, Texas), sc-74569), rabbit anti-γH2AX (p-Histone H2A.X (ser 139), 0.13 µg/ml, Santa Cruz, sc-101696), rabbit anti-DMC1 (Dmc1 (H-100), 8 µg/ml, Santa Cruz, sc-22768). Those used for histology are as follows: mouse anti-SYCP3 (SCP-3 (D-1), 1 µg/ml, Santa Cruz, sc-74569), goat anti-YTHDC2 (YTHDC2 (G-19), 5 µg/ml, Santa Cruz, sc-249370), rabbit anti-YTHDC2 (YTHDC2, 1 µg/ml, Bethyl Laboratories (Montgomery, Texas), A303-025A), rabbit anti-CCNA2 (anti-Cyclin A2 (Y193), 2.5 µg/ml, Abcam (Cambridge, Massachusetts), ab32386), mouse anti-α-tubulin (anti-α-Tubulin, 2.5 µg/ml, Millipore (Billerica, Massachusetts), MABT205), rabbit anti-pH3 (anti-phospho-Histone H3 (Ser10), 1 µg/ml, Upstate (Millipore), 06-570), mouse anti-BrdU (Anti-Bromodeoxyuridine, 1 µg/ml, Roche (Mannheim, Germany), 1170376).

## RNA-seq

Animals for RNA-seq were derived from crosses between heterozygote carriers after five or six backcrosses to B6 mice. Sequencing was performed with the following animals: three 8 *dpp*-old *Ythdc2^em1/em1^* mutant and wild-type littermates; two 9 *dpp*-old *Ythdc2^em1/em1^* mutant and wild-type littermates and one 9 *dpp*-old *Ythdc2^em1/em1^* mutant and heterozygous littermates; two 10 *dpp*-old *Ythdc2^em1/em1^* mutant and heterozygous littermates. The six 8 and 9 *dpp*-old mutant and wild-type or heterozygous pairs were from six independent litters. The two 10 *dpp*-old mutant and heterozygous pairs were from a seventh litter.

Total RNA from a single testis per animal, after removing the tunica albuginea, was extracted using the RNeasy Plus Mini Kit containing a genomic DNA eliminator column (Qiagen, Germantown, Maryland), according to manufacturers' instructions. Sequencing was performed at the Integrated Genomics Operation of MSKCC. 500 µg of total RNA underwent polyA capture and library preparation using the Truseq Stranded Total RNA library preparation chemistry (Illumina). Sequencing was performed using the Illumina HiSeq platform (2 × 50 bp paired-end reads) to generate, on average, 44 million paired reads per sample.

Resulting RNA-seq fastq files were aligned using STAR version 2.5.3a (*Dobin et al., 2013*) to the mouse genome (GRCm38/mm10) using Gencode M11 transcriptome assembly (*Mudge and Harrow, 2015*) for junction points. RNAseq counts were annotated by using subread version 1.4.5-p1 featureCounts (*Liao et al., 2014*). Counts were normalized to transcripts per kilobase million (TPM) values (*Li et al., 2010*) for plotting. Differentially expressed genes were calculated using DESeq2 (*Love et al., 2014*). Hierarchical clustering was performed on z-scaled TPM values of differentially expressed genes in *Figure 6B* using Manhattan distance and Ward's clustering method and the dendrogram was cut to give four clusters.

RNA-seq data for purified testicular cell types are published (GEO GSE43717 [*Soumillon et al., 2013*]). Fragments per kilobase per million mapped reads (FPKM) values, provided by the authors, were matched to our RNA-seq data by Ensembl Gene ID and used in *Figure 6C* for all differentially expressed genes belonging to Cluster I or Clusters II and III, except *Pla2g4b* and *Ankrd31*, which were absent from the published data. Enriched gene ontology (GO) terms belonging to the biological process sub-ontology were identified with DAVID using the BP_DIRECT category (*Huang et al., 2009b*; *a*) with a cutoff of Benjamini-Hochberg-adjusted p-value<0.05. m$^6$A sequencing data for 8.5 *dpp*-old wild-type mice are published (GEO GSE98085 [*Hsu et al., 2017*]). The list of genes containing m$^6$A peaks, provided by the authors, was matched to our RNA-seq data by gene name and used in *Figure 6E*.

## Protein domain and structure analysis

Domain annotations were obtained from SMART (*Letunic et al., 2015*) and pfam database (*Finn et al., 2016*) searches. Atomic coordinates of NMR and crystal structures of YTH domains were retrieved from the following Protein Data Bank (PDB) entries: 2YU6 (human YTHDC2 apo structure), 4R3I (RNA-bound human YTHDC1), 5DNO (RNA-bound *S. pombe* Mmi1), and 5H8A (*S. pombe* Mmi1 apo structure). Alignments of three-dimensional structures were performed using cealign (*Vertrees, 2007*) and the figures prepared using PyMol (*Schrodinger, LLC, 2015*). Protein accession numbers are listed in *Supplementary files 2* and *3*.

## Expression, purification and biochemical characterization of recombinant YTHDC2

The mouse YTHDC2 cDNA sequence was codon-optimized for baculovirus insect cell expression and cloned into pFastbac-1 with a TEV-cleavable C-terminal decahistidine tag or an N-terminal MBP tag and a C-terminal tandem hexahistidine-3×FLAG tag. Bacmids were prepared by transforming chemically competent DH10bac cells with the mouse YTHDC2 plasmid construct. Baculoviruses were generated in *Spodoptera frugiperda* Sf9 cells using the Bac-to-Bac Baculovirus Expression System (Thermo Fisher Scientific, Waltham, Massachusetts) following the manufacturer's protocol. High Five cells were grown in Sf-900 II serum free media to a density of $2 \times 10^6$ cells per ml and infected with the P3 viral stock at a multiplicity of infection of 2. Cells were incubated with shaking at 27°C for the next 48 hr prior to harvesting. Cells were resuspended in 50 mM Tris-HCl pH 8, 500 mM NaCl, 0.2% v/v Tween 20, 2 mM β-mercaptoethanol and 1× Complete protease inhibitor tablet (Roche), lysed by sonication, and centrifuged at 40,000 ×g for 20 min. The supernatant was collected and incubated with $Ni^{2+}$-NTA resin for at least 30 min. The resin was washed three times with 10 volumes of wash buffer containing 25 mM Tris-HCl pH 8, 500 mM NaCl, 0.01% Tween 20, 1 mM PMSF and 10 mM imidazole. His-tagged proteins were eluted in 25 mM Tris-HCl pH 8, 100 mM NaCl, 1 mM PMSF and 250 mM imidazole, filtered, and further purified using a Heparin HP column (GE Healthcare, Chicago, Illinois). Fractions containing the recombinant protein were pooled, treated with TEV protease overnight, and passed through a HisTrap HP column (GE Healthcare) to remove uncleaved proteins. Finally, the protein was subjected to gel filtration chromatography on a Superdex 200 10/300 GL column (GE Healthcare) in 20 mM Tris-HCl pH 8, 350 mM NaCl and 0.1 mM tris(2-carboxyethyl)phosphine (TCEP). Protein-containing fractions were pooled, concentrated, flash frozen, and stored at −80°C. MBP-tagged proteins were obtained by $Ni^{2+}$-affinity and gel filtration chromatography as described above except the tags were not cleaved. Protein concentrations were determined using the Bradford protein assay (Bio-Rad, Hercules, California).

Duplex RNA substrates for helicase assays were prepared by mixing stoichiometric ratios of each strand (*Supplementary file 4*) in a buffer containing 20 mM Tris-HCl pH 7.0 and 100 mM potassium acetate followed by heating to 95°C for 5 min and incubation at 16°C for at least one hour. RNA unwinding reactions were carried out at 22°C in 40 mM MOPS-NaOH pH 6.5, 50 mM NaCl, 0.5 mM $MgCl_2$, 5% v/v glycerol, 5 mM β-mercaptoethanol and 0.5 U/µl RNase inhibitor (Murine, NEB). MBP-tagged proteins were assayed at 30°C in 25 mM Tris-HCl pH 8.0, 50 mM NaCl, 2 mM $MgCl_2$, 5 mM β-mercaptoethanol and 0.8 U/µl RNase inhibitor (Murine, NEB). RNA substrates (10 nM) were pre-incubated with the protein for 5 min. Reactions were initiated by simultaneous addition of 2 mM ATP (pH 7), 2 mM $MgCl_2$ and 400 nM DNA trap. Aliquots were quenched with a solution containing 0.5% SDS, 5 mM EDTA and 25% v/v glycerol. Samples were subjected to native PAGE using a 20% Novex TBE gel (Thermo Fisher Scientific). Gel bands were imaged using a Typhoon FLA 9500 laser scanner (GE Healthcare) and quantified using ImageJ (*Schneider et al., 2012*). The fraction unwound was calculated from the ratio of the band intensity of the unwound substrate to the sum of the intensities of the unreacted substrate and the product. Data were analyzed using Graphpad Prism version 7. Observed rate constants ($k_{obs}$) were obtained using the pseudo-first order rate law: fraction unwound = $A(1-e^{kobs*t})$ where A is the reaction amplitude and t is time (*Jankowsky et al., 2000*).

## Phylogenetic analysis

Mouse YTHDC2 was used as a query in searches using BLASTP (version 2.6.1 [*Altschul et al., 1997*]) and CDART (conserved domain architecture retrieval tool [*Geer et al., 2002*]) using NCBI servers. Searches were performed iteratively, with new searches seeded with hits from initial searches. When multiple accessions were present in the same species, we chose the longest isoform available. Searches were further repeated in targeted mode (i.e., restricting the taxon ID in BLASTP) to examine specific lineages in more detail (e.g., Nematoda, Schizophora, Insecta, Crustacea). MegAlign Pro (DNASTAR Inc., Madison, Wisconsin) version 14.1.0 (118) was used to generate multiple sequence alignments with Clustal Omega (*Sievers et al., 2011*) or MUSCLE (*Edgar, 2004*) using default settings; to calculate alignment distances using the scoredist function (*Sonnhammer and Hollich, 2005*); and to output neighbor-joining trees using the BioNJ algorithm (*Gascuel, 1997*). COBALT (*Papadopoulos and Agarwala, 2007*) alignments were carried out on the NCBI server (https://www.ncbi.nlm.nih.gov/tools/cobalt/re_cobalt.cgi). NCBI taxonomic cladograms were constructed using

the PhyloT web tool (http://phylot.biobyte.de/). Trees were visualized using the interactive tree of life (ITOL) server (http://itol.embl.de/) (*Letunic and Bork, 2016*) or FigTree version 1.4.3 (http://tree.bio.ed.ac.uk/software/figtree).

From targeted BLASTP searches of the following taxa, no clear matches to the YTHDC2 architecture were observed: Amoebozoa, Fornicata, Euglenozoa, Alveolata, Apusozoa, Cryptophyta, Haptophyceae, Heterolobosea, Parabasalia, Rhizaria, Rhodophyta, Stramenopiles. In these lineages, the closest homologs found were RNA helicase-like proteins more similar in architecture to the DHX30 family, without R3H and YTH domains and lacking the ARD insertion between the helicase core domains.

We also did not find clear matches to the YTHDC2 architecture in fungi. The closest *Saccharomyces cerevisiae* homolog (YLR419W) lacks the diagnostic ARD insertion between the helicase core domains, has N-terminal UBA (ubiquitin-associated) and RWD domains rather than an R3H domain, and lacks a YTH domain. YLR419W thus more closely resembles human DHX57 (*Figure 8—figure supplement 1A*), which is indeed the top hit when YLR419W is used as the query in a BLASTP search against the human genome (GenBank accession AAH65278.1; 29% identity, E-value $2 \times 10^{-110}$).

## Data availability

Reagents and mouse strains are available upon request. RNA-seq data are available at Gene Expression Omnibus (GEO) with the accession number: GSE108044.

## Acknowledgements

We thank Keeney lab members Corentin Claeys Bouuaert for initial expression and purification of recombinant YTHDC2; Shintaro Yamada for advice on RNA-seq analysis; Laurent Acquaviva for help with chromosome spreads; and Luis Torres, Diana Y Eng, and Jacquelyn Song for assistance with genotyping and mouse husbandry. We thank Ning Fan, Dmitry Yarilin, Afsar Barlas, Yevgeniy Romin, Sho Fujisawa, Elvin Feng, Matt Brendel and Mesruh Turkekul (MSKCC Cytology Core Facility) for help with histological analyses and imaging. We thank Peter Romanienko (MSKCC Mouse Genetics Core Facility) for designing the gRNA. We thank Julie White (MSKCC Laboratory of Comparative Pathology Core Facility) for anatomic pathology. We thank the Genetic Analysis Facility (Centre for Applied Genomics, Hospital for Sick Children, Toronto, ON, Canada) for microarray analysis; and the MSKCC Mouse Genetics Core Facility for CRISPR/Cas9-targeted mice. For RNA-seq and whole-exome sequencing, we thank the MSKCC Integrated Genomics Operation. We are grateful to Gabriel Livera and Jeremy Wang for sharing unpublished information. We acknowledge the RIKEN Structural Genomics/Proteomics Initiative for providing the YTHDC2 YTH domain solution structure. We also acknowledge the ENCODE Consortium (*ENCODE Project Consortium, 2012*) and the ENCODE production laboratory of Thomas Gingeras (Cold Spring Harbor Laboratory) for generating the ENCODE datasets used in this manuscript.

## Additional information

### Funding

| Funder | Grant reference number | Author |
|---|---|---|
| Howard Hughes Medical Institute | | M Rhyan Puno<br>Christopher D Lima<br>Scott Keeney |
| Eunice Kennedy Shriver National Institute of Child Health and Human Development | R37 HD035455 | Kathryn V Anderson |
| Human Frontier Science Program | | Devanshi Jain |
| National Cancer Institute | P30 CA008748 | Nathalie Lailler |
| National Institute of General Medical Sciences | R35 GM118080 | M Rhyan Puno<br>Christopher D Lima |

| | | |
|---|---|---|
| Starr Cancer Consortium | I9-A9-071 | Cem Meydan<br>Christopher E Mason |
| Bert L and N Kuggie Vallee Foundation | | Cem Meydan<br>Christopher E Mason |
| WorldQuant Foundation | | Cem Meydan<br>Christopher E Mason |
| Pershing Square Sohn Cancer Research Alliance | | Cem Meydan<br>Christopher E Mason |
| National Aeronautics and Space Administration | NNX14AH50G 15-15Omni2-0063 | Cem Meydan<br>Christopher E Mason |
| Bill and Melinda Gates Foundation | OPP1151054 | Cem Meydan<br>Christopher E Mason |
| Cycle for Survival | | Nathalie Lailler |
| Marie-Josée and Henry R. Kravis Center for Molecular Oncology | | Nathalie Lailler |

The funders had no role in study design, data collection and interpretation, or the decision to submit the work for publication.

### Author contributions

Devanshi Jain, Conceptualization, Formal analysis, Funding acquisition, Investigation, Writing—original draft, Writing—review and editing; M Rhyan Puno, Formal analysis, Writing—review and editing, Investigation, Writing—original draft; Cem Meydan, Formal analysis, Writing—review and editing; Nathalie Lailler, Formal analysis, Analysis of exome sequencing data to map *ketu* mutation; Christopher E Mason, Christopher D Lima, Kathryn V Anderson, Supervision, Funding acquisition, Writing—review and editing; Scott Keeney, Conceptualization, Formal analysis, Supervision, Funding acquisition, Writing—original draft, Writing—review and editing

### Author ORCIDs

Devanshi Jain (iD) http://orcid.org/0000-0002-5027-0549
Christopher D Lima (iD) http://orcid.org/0000-0002-9163-6092
Scott Keeney (iD) http://orcid.org/0000-0002-1283-6417

### Ethics

Animal experimentation: This study was performed in strict accordance with the recommendations in the Guide for the Care and Use of Laboratory Animals of the National Institutes of Health. All experiments conformed to regulatory standards and were approved by the Memorial Sloan Kettering Cancer Center Institutional Animal Care and Use Committee under protocol #01-03-007.

### Decision letter and Author response

Decision letter https://doi.org/10.7554/eLife.30919.048
Author response https://doi.org/10.7554/eLife.30919.049

## Additional files

### Supplementary files

• Supplementary file 1. Expression fold changes of differentially expressed genes in *Figure 6B*.
DOI: https://doi.org/10.7554/eLife.30919.036

• Supplementary file 2. Protein accession numbers for *Figure 8A*.
DOI: https://doi.org/10.7554/eLife.30919.037

• Supplementary file 3. Protein accession numbers for *Figures 8B–D*, *10* and *11*.
DOI: https://doi.org/10.7554/eLife.30919.038

• Supplementary file 4. Genotyping primers and oligos used to make helicase assay substrates.

DOI: https://doi.org/10.7554/eLife.30919.039

• Transparent reporting form
DOI: https://doi.org/10.7554/eLife.30919.040

## Major datasets

The following dataset was generated:

| Author(s) | Year | Dataset title | Dataset URL | Database, license, and accessibility information |
|---|---|---|---|---|
| Jain D, Puno R, Meydan C, Lailler N, Mason CE, Lima CD, Anderson KV, Keeney S | 2017 | ketu mutant mice uncover an essential meiotic function for the ancient RNA helicase YTHDC2 | https://www.ncbi.nlm.nih.gov/geo/query/acc.cgi?acc=GSE108044 | Publicly available at the NCBI Gene Expression Omnibus (accession no: GSE108044) |

The following previously published datasets were used:

| Author(s) | Year | Dataset title | Dataset URL | Database, license, and accessibility information |
|---|---|---|---|---|
| Hsu PJ, Zhu Y, Ma H, Cui Y, Shi X, Lu Z, Shi H, Luo G, Dai Q, Shen B, He C | 2017 | YTHDC2 regulates spermatogenesis through promoting the translation of N6-methyladenosine-modified RNA | https://www.ncbi.nlm.nih.gov/geo/query/acc.cgi?acc=GSE98085 | Publicly available at the NCBI Gene Expression Omnibus (accession no: GSE98085) |
| Soumillon M, Necsulea A, Weier M, Brawand D, Zhang X, Gu H, Barthès P, Kokkinaki M, Nef S, Gnirke A, Dym M, de Massy B, Mikkelsen TS, Kaessmann H | 2013 | Cellular source and mechanisms of high transcriptome complexity in the mammalian testis (RNA-Seq cells) | https://www.ncbi.nlm.nih.gov/geo/query/acc.cgi?acc=GSE43717 | Publicly available at the NCBI Gene Expression Omnibus (accession no: GSE43717) |

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
