## [Decision Letter]

Thank you for submitting your article "*ketu* mutant mice uncover an essential meiotic function for the ancient, putative RNA helicase YTHDC2" for consideration by *eLife*. Your article has been favorably evaluated by Kevin Struhl (Senior Editor) and three reviewers, one of whom, Bernard de Massy, is a member of our Board of Reviewing Editors. The following individual involved in review of your submission has agreed to reveal their identity: R Scott Hawley (Reviewer #2).

The reviewers have discussed the reviews with one another and the Reviewing Editor has drafted this decision to help you prepare a revised submission.

Summary:

You have performed an interesting study, starting with a genetic screen for meiotic mutants, leading to identification and analysis of *Ythdc2* in mice. This analysis demonstrates an essential role of *Ythdc2* for meiosis with an altered progression during meiotic prophase in mutant spermatocytes. You also show that *Ythdc2* localizes to the cytoplasm of spermatocytes. The phylogeny of *Ythdc2* reveals conservation and divergence among orthologs with loss of domains. Of specific interest is the YTH domain present in most metazoans. Based on published structures, the YTH domain of *Ythdc2* is predicted to bind to m6A-RNA. Interestingly the authors detect similarities between Meioc, a partner of *Ythdc2* and *D. melanogaster* Bam, a partner of Bgcn, a paralogue of *Ythdc2*. This leads to the proposal of an evolutionary conserved function for the *Ythdc2*/Meioc complex.

Essential revisions:

With respect to previous studies on *Ythdc2/Meioc* (Xu et al. 2015; Abby et al. 2016; Soh et al. 2017), this paper provides support for *Ythdc2* and *Meioc* acting as a complex and important for meiosis progression, a concept already proposed by others. However, the questions raised in previous studies on *Meioc*, specifically which process is altered in meiotic prophase and how remain unsolved. In particular this paper provides no advance with the central question which is to understand how these genes impact on RNA metabolism. The proposed activity of this complex on RNA stability and/or translation on a subset of genes remains unknown. In order for this study to provide a significant advance in the understanding of *Ythdc2* and for publication in *eLife*, at least two levels of information are absolutely required: How similar are the *Meioc* and *Ythdc2* phenotypes? And what are the targets of *Ythdc2*/how does *Ythdc2* impact RNA metabolism? The recent paper by Hsu et al., 2017 does not shed light on these questions.

First, it must be made clear how similar *Meioc* and *Ythdc2* phenotypes are: in order to provide an extensive comparison with *Meioc* mutants, you need to phenotypes in testis before 10 *dpp*, evaluate meiotic S phase progression, bouquet formation, DSB repair activity (besides gH2AX) and transcriptome analysis.

Second, you have an opportunity to clarify the interaction between *Ythdc2* and specific RNAs (by Clip-Seq for instance or other approaches) as previously attempted, but by including the *Ytcdh2* mutant as negative control. This is important, since no information is provided about the specificity of the commercial anti-*Ythdc2* antibodies used for such experiments.

With this additional information, although descriptive, this paper should be able to strengthen the proposition of *Meioc/Ythdc2* acting as a complex and to provide a first step into the predicted molecular activity of *Ythdc2*. Additional results from approaches to determine the mode of action of the putative *Ythdc2/Meioc* complex, i.e., on RNA stability and/or translation would be interesting, and you are encouraged to add them, but they are not absolutely required.

---

## [Author Response]

Essential revisions:With respect to previous studies on Ythdc2/Meioc (Xu et al. 2015; Abby et al. 2016; Soh et al. 2017), this paper provides support for Ythdc2 and Meioc acting as a complex and important for meiosis progression, a concept already proposed by others. However, the questions raised in previous studies on Meioc, specifically which process is altered in meiotic prophase and how remain unsolved. In particular this paper provides no advance with the central question which is to understand how these genes impact on RNA metabolism. The proposed activity of this complex on RNA stability and/or translation on a subset of genes remains unknown. In order for this study to provide a significant advance in the understanding of Ythdc2 and for publication in eLife, at least two levels of information are absolutely required: How similar are the Meioc and Ythdc2 phenotypes? And what are the targets of Ythdc2/how does Ythdc2 impact RNA metabolism? The recent paper by Hsu et al., 2017 does not shed light on these questions.First, it must be made clear how similar Meioc and Ythdc2 phenotypes are: in order to provide an extensive comparison with Meioc mutants, you need to phenotypes in testis before 10 dpp, evaluate meiotic S phase progression, bouquet formation, DSB repair activity (besides gH2AX) and transcriptome analysis.

We agree that this is an important point, and we welcome the opportunity to strengthen the conclusion that YTHDC2 and MEIOC function together in germ cells. To address this, we have incorporated extensive new data into the paper:

- We added phenotypic analysis of 8 *dpp* males and more extensive analysis at 10 and 12 *dpp* (Figure 3, Figure 3—figure supplement 1, Figure 5, Figure 5—figure supplement 1).

- We performed transcriptomic (RNA-seq) analysis of testes at 8, 9, and 10 *dpp* (Figure 6 and Figure 6—figure supplement 1).

- We evaluated meiotic S phase progression by BrdU pulse labeling (Figure 5 and Figure 5—figure supplement 1).

- We performed DMC1 staining (Figure 1).

- We extended the analysis quantitatively and over several ages of CCNA2 persistence and of premature metaphase in spermatocytes (Figure 4, Figure 4—figure supplement 1, Figure 5, Figure 5—figure supplement 1).

Collectively, these findings strengthen and confirm the conclusion that *Meioc* and *Ythdc2* mutant phenotypes in the male germline are highly similar if not identical, which in turn supports the hypothesis that the two proteins function together as a complex. This is also supported by complementary experiments by the Fuller and Pillai labs, which were reported while we were preparing the revision of our manuscript (Bailey et al., 2017 and Wojtas et al., 2017). We also point out that the phylogenetic analysis (an aspect wholly unique to our paper) also makes a strong case for functional collaboration of the two proteins, as it is well established in *D. melanogaster* that Bgcn and Bam function together in the germline. We elected not to evaluate bouquet formation as explained further below.

Second, you have an opportunity to clarify the interaction between Ythdc2 and specific RNAs (by Clip-Seq for instance or other approaches) as previously attempted, but by including the Ytcdh2 mutant as negative control. This is important, since no information is provided about the specificity of the commercial anti-Ythdc2 antibodies used for such experiments.

As we discussed with the editor, we agree this is interesting to do, but we felt it would be redundant with information in the Fuller paper. Instead, we proposed to provide biochemical characterization of YTHDC2 helicase activity. We demonstrate 3’ to 5’ ATP-dependent RNA helicase activity, and show that the *ketu* mutation most likely affects folding and/or stability of the protein (Figure 9 and Figure 9—figure supplement 1). The paper by Wojtas et al. (which we were not aware of at the time of our discussions with the editor) also examined helicase activity, but our findings go beyond theirs by establishing information about substrate specificity for the unwinding activity, and we evaluate the effect of the *ketu* mutation.

With this additional information, although descriptive, this paper should be able to strengthen the proposition of Meioc/Ythdc2 acting as a complex and to provide a first step into the predicted molecular activity of Ythdc2. Additional results from approaches to determine the mode of action of the putative Ythdc2/Meioc complex, i.e., on RNA stability and/or translation would be interesting, and you are encouraged to add them, but they are not absolutely required.

We agree that it remains an important issue what the function and biochemical mechanism of YTHDC2 is in post-transcriptional control of gene expression: Is it principally involved in translation control, RNA stability, or both? What are the relevant direct RNA targets? How does helicase activity contribute to its function? However, we point out that none of the six papers on YTHDC2 and MEIOC (including ours) has yet succeeded in definitively answering these points because they are very challenging to dissect. Multiple hypotheses have been proposed, and our paper complements and significantly extends the other papers on this front. Our RNA-seq analysis of multiple time points during the germline transition into meiosis provides a novel view of the effect of YTHDC2 loss on transcript abundance, demonstrating that there are polar opposite effects on transcript levels for different classes of developmentally regulated genes. These findings help reconcile confusing and contradictory claims in some of the other *Meioc* and *Ythdc2* papers. We also point out that our paper is the only one to extensively analyze the remarkable phylogenetic behavior of these genes, thus we are the first to place the more mechanistic findings into a broader evolutionary context.